# Zero-Shot Generalization of GNNs over Distinct Attribute Domains

**Yangyi Shen** [1]  **Jincheng Zhou** [2]  **Beatrice Bevilacqua** [2]  **Joshua Robinson** [1]  **Charilaos Kanatsoulis** [1]
**Jure Leskovec** [1]  **Bruno Ribeiro** [2]

## Abstract

Traditional Graph Neural Networks (GNNs) cannot generalize to new graphs with node attributes different from the training ones, making zero-shot generalization across different node attribute domains an open challenge in graph machine learning. In this paper, we propose STAGE, which encodes *statistical dependencies* between attributes rather than individual attribute values, which may differ in test graphs. By assuming these dependencies remain invariant under changes in node attributes, STAGE achieves provable generalization guarantees for a family of domain shifts. Empirically, STAGE demonstrates strong zero-shot performance on medium-sized datasets: when trained on multiple graph datasets with different attribute spaces (varying in types and number) and evaluated on graphs with entirely new attributes, STAGE achieves a relative improvement in Hits@1 between 40% to 103% in link prediction and a 10% improvement in node classification compared to state-of-the-art baselines.

## 1. Introduction

Zero-shot generalization refers to the ability of the model to handle unseen test data without additional training or adaptation (Larochelle et al., 2008; Xian et al., 2017; Wang et al., 2022). An essential prerequisite for zero-shot generalization is a unified input space where models can learn and transfer prediction patterns across domains. While this challenge has been addressed in areas like natural language through tokenization techniques that represent any text through a fixed vocabulary (Samuel & Øvrelid, 2023), graphs present unique challenges in achieving such unified input space.

Attributes in graphs can vary significantly across domains.

Node attributes in test graphs can differ from those in training graphs in four key ways: (1) their types (e.g., continuous vs. categorical variables); (2) their names (e.g., *RAM* specifications in ecommerce graphs and clothing *size* in retail graphs, as illustrated in Figure 1); (3) their semantics, where attributes with the same name can have different meanings across domains – for instance, the meaning of *size* differs substantially between electronics and clothing domains; (4) their cardinality, as graphs may contain varying numbers of node attributes. *These challenges make it difficult to define a unified input space that enables zero-shot generalization to unseen attributed graphs.*

For these reasons, training graph models that can zero-shot generalize to new graphs with unseen attribute domains remains an open challenge. Recent approaches address this problem using various strategies. One approach is to ignore node attributes to focus solely on graph topology, but this strategy may be leaving valuable node attribute information unutilized. Another line of work seeks to unify input spaces by converting graphs and attributes into text representations, which are then processed by pretrained text encoders (Chen et al., 2024a; Huang et al., 2023; Liu et al., 2024; Zhang et al., 2023). While promising, these approaches may struggle with numerical attributes (Collins et al., 2024; Gruver et al., 2024; Schwartz et al., 2024). Recently, Zhao et al. (2024b) proposed an analytical approach for making predictions on new graphs with potentially new attributes. However, this approach sidesteps the fundamental challenge of creating a unified input space.

In this paper, we introduce STAGE (**S**tatistical **T**ransfer for **A**ttributed **G**raph **E**mbeddings), which transforms node attributes from their "absolute" natural space into a *relative* space that captures statistical dependencies between attributes. For instance, as illustrated in Figure 1, these dependencies manifest themselves as correlations driving purchases across domains, which remain invariant even when the purchased items and their attributes change. In practice, STAGE represents such statistical dependencies through a two-step process that transforms node attributes into fixed-dimensional edge embeddings, achieving a unified input space alongside provable invariance to changes in attribute values (including their types, names and semantics), as well as to permutations of attribute order and permuta-

---

[1]Department of Computer Science, Stanford University, Stanford, USA [2]Department of Computer Science, Purdue University, West Lafayette, USA. Correspondence to: Yangyi Shen <pyyshen@stanford.edu>.

*Proceedings of the $42^{nd}$ International Conference on Machine Learning*, Vancouver, Canada. PMLR 267, 2025. Copyright 2025 by the author(s).

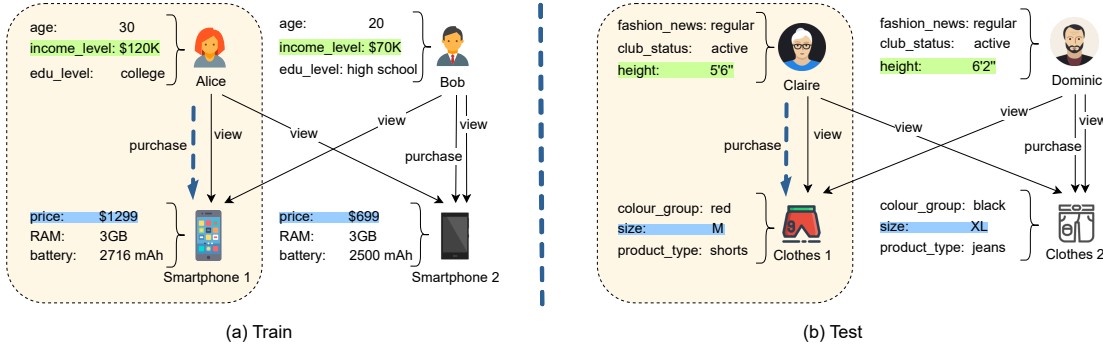

Figure 1: The task of zero-shot generalization to attributed graphs with unseen attributes. Attributes in test are different than those in train in types and semantics, but attributes associated with an edge are highly correlated in both train and test (e.g. income level is positively correlated to phone price in (a) and height is positively correlated to size in (b)). Our STAGE learns these *statistical dependencies* among attributes to perform zero-shot transfer across distinct attribute domains.

tions of node identities. Specifically, STAGE first constructs a weighted *STAGE-edge-graph* for each edge in the input graph, where the nodes represent attributes of the edge endpoints and the edge weights capture dependencies between the attributes. Then, STAGE uses an additional shallow GNN to generate embeddings for each *STAGE-edge-graph*. Finally, STAGE applies the original GNN to a modified input graph, which contains only the newly generated edge embeddings but not the node attributes.

The complexity of STAGE is linear in the size of the input graph and quadratic in the number of attributes, as it captures pairwise statistical dependencies between attributes over the edges of the graph. This makes STAGE particularly well-suited for small to medium-sized datasets, where it strikes a balance between computational feasibility and strong generalization performance.

We prove that STAGE can learn domain-independent representations for certain types of domain shifts, enabling zero-shot generalization. Experimentally, for link prediction in e-commerce networks spanning six distinct product domains, STAGE achieves up to 103% improvement in Hits@1 compared to the strongest baseline. In node classification tasks on social networks, STAGE achieves approximately 10% better performance than the strongest baseline.

## 2. STAGE

Let $G = (V, E, \boldsymbol{X})$ an attributed graph, where $V$ is the set of nodes, $E$ the set of edges, and $\boldsymbol{X} = \{\boldsymbol{x}^v\}_{v \in V}$ the set of node attributes $\boldsymbol{x}^v$ for each node $v \in V$. We assume that all $\boldsymbol{x}^v$ belong to some measurable space of dimension $d \geq 1$.

To design a model capable of generalizing to test graphs that may have node attributes living in a different space than $\boldsymbol{X}$, we propose a projection map that transforms the node attributes $(\boldsymbol{x}^u, \boldsymbol{x}^v)$ of the endpoints of an edge $(u, v) \in E$

into a fixed-dimensional pairwise embedding

$$\mathcal{P} : (\boldsymbol{x}^u, \boldsymbol{x}^v) \mapsto \boldsymbol{r}^{uv} \in \mathbb{R}^k, \quad k \geq 1. \quad (1)$$

By using pairwise embeddings, STAGE can *model relationships between attributes belonging to different nodes*. For instance, it can capture the relation between the attributes of the customer node Alice and the attributes of the product node Phone1 in Figure 2(a), such as the correlation between income level and price. We design the mapping $\mathcal{P}$ by building a graph based on the pairwise *pdf* attribute descriptors. Viewing node attributes through their *pdf*s maps potentially non-aligned node attribute spaces into a universal space of densities, enabling consistency across diverse domains. The modeling of probabilities generalizes the learning of rules like "people with higher income level tend to buy expensive phones,' to abstract relationships like "high values in $X_1$ correlate with high values in $X_2$", enabling knowledge transfer across domains with different attributes.

Concretely, let A and B be a random pair of nodes jointly and uniformly sampled from the edge set, $(A, B) \sim \text{Unif}(E)$. Let $\text{x}_i^A$ denote the random variable of the $i$-th attribute value of random node A, and $\text{x}_j^B$ the $j$-th attribute value of random node B. Given a specific pair of distinct nodes $u, v \in V$ and specific attribute values $x_i^u$ and $x_j^v$, we define $p(x_i^u | x_j^v)$ from the conditional probabilities as follows, accounting for mixture of totally ordered (e.g., scalar) and unordered (e.g., categorical) attributes[1]:

- $p(x_i^u | x_j^v) := \mathbb{P}(\text{x}_i^A \leq x_i^u | \text{x}_j^B \leq x_j^v)$, if both attribute $i$ and $j$ are totally ordered.

- $p(x_i^u | x_j^v) := \mathbb{P}(\text{x}_i^A = x_i^u | \text{x}_j^B \leq x_j^v)$, if attribute $i$ is unordered and attribute $j$ is totally ordered.

- $p(x_i^u | x_j^v) := \mathbb{P}(\text{x}_i^A \leq x_i^u | \text{x}_j^B = x_j^v)$, if attribute $i$ is totally ordered and attribute $j$ is unordered.

---

[1]For brevity we omit the distribution $(A, B) \sim \text{Unif}(E)$, writing $\mathbb{P}$ instead of $\mathbb{P}_{(A,B) \sim \text{Unif}(E)}$ from now on.

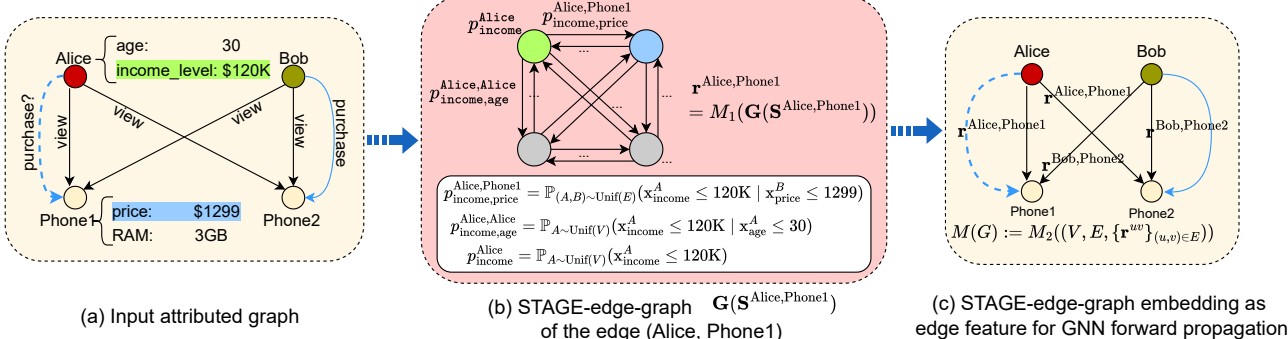

(a) Input attributed graph

(b) STAGE-edge-graph $\mathbf{G}(\mathbf{S}^{\text{Alice,Phone1}})$
of the edge (Alice, Phone1)

(c) STAGE-edge-graph embedding as
edge feature for GNN forward propagation

Figure 2: Given an input attributed graph $G$ (a), STAGE builds a *STAGE-edge-graph* (b) for every edge in $G$. Nodes in a STAGE-edge-graph correspond to attributes of the two edge endpoints, and the node and edge attributes are the empirical marginal and conditional probabilities of attribute values (Equations (2) and (3)). STAGE applies the intra-edge GNN on STAGE-edge-graphs (b) to obtain an edge embedding for each input graph edge, and then applies the inter-edge GNN on the modified graph containing these edge embeddings but not the node attributes (c). Details are provided in Algorithms 1 and 2.

- $p(x_i^u | x_j^v) := \mathbb{P}(\mathbf{x}_i^A = x_i^u | \mathbf{x}_j^B = x_j^v)$, if both attribute $i$ and $j$ are unordered.

If $u = v$, we change the sampling distribution to (A) $\sim$ Unif$(V)$ and let B = A so that STAGE can also model dependencies between attributes of the same node. If $i = j$, we change the conditional probability to $p(x_i^u) := \mathbb{P}(\mathbf{x}_i^A = x_i^u)$ if attribute $i$ is unordered and $p(x_i^u) := \mathbb{P}(\mathbf{x}_i^A \leq x_i^u)$ if attribute $i$ is totally ordered. This allows STAGE to also model each attribute independently through its *pdf* or *cdf*.

In practice, these probabilities can be empirically estimated from the input data. For the node-pair $u, v$ we define a conditional probability matrix $\mathbf{S}^{uv}$, with indices $i, j \in \{1, \ldots, 2d\}, i \neq j$, organized such that indices 1 to $d$ correspond to attributes of node $u$ and indices $d + 1$ to $2d$ correspond to attributes of node $v$:

$$\mathbf{S}_{ij}^{uv} = \begin{cases} p(x_i^u \mid x_j^u) & \text{if } i \leq d \text{ and } j \leq d, \\ p(x_{i-d}^v \mid x_{j-d}^v) & \text{if } d < i \leq 2d \text{ and } d < j \leq 2d, \\ p(x_i^u \mid x_{j-d}^v) & \text{if } i \leq d \text{ and } d < j \leq 2d, \\ p(x_{i-d}^v \mid x_j^u) & \text{if } d < i \leq 2d \text{ and } j \leq d. \end{cases} \tag{2}$$

and for the diagonal $i = j$ we define,

$$\mathbf{S}_{ij}^{uv} = \begin{cases} p(x_i^u) & \text{if } i \leq d, \\ p(x_i^v) & \text{if } i > d, \end{cases} \tag{3}$$

The matrix $\mathbf{S}^{uv}$ is the core node-pair data representation STAGE uses. This matrix is used to define a graph structure which we call a STAGE-edge-graph, illustrated in Figure 2(b), which captures, for the pair of nodes $u$ and $v$, the interactions among all pairs of attributes.

**Definition 2.1** (STAGE-edge-graph). Given a pair of nodes $u, v \in V$, a STAGE-edge-graph for $(u, v)$ is a fully connected, weighted, directed graph $\mathbf{G}(\mathbf{S}^{uv})$ with $2d$ nodes, where node $i$ has a scalar attribute $\mathbf{S}_{ii}^{uv}$, and edge $(i, j)$ has a scalar attribute $\mathbf{S}_{ij}^{uv}$.

**STAGE algorithm.** As illustrated in Figures 2(b) and 2(c), STAGE uses a STAGE-edge-graph for each edge in the input graph in a two-stage process to produce attribute-domain-transferable representations. First, STAGE uses a GNN to obtain embeddings for each STAGE-edge-graph. These edge embeddings replace the original node attributes, resulting in a modified graph which is fed into a second GNN to solve the overall task, producing node, link, or graph representation. The two steps of STAGE are as follows:

1. *(Intra-edge)* Each $\mathbf{G}(\mathbf{S}^{uv})$ is processed with a GNN $M_1$ to produce edge-level embeddings $\boldsymbol{r}^{uv} = M_1(\mathbf{G}(\mathbf{S}^{uv}))$.

2. *(Inter-edge)* A second GNN $M_2$ processes the modified graph $G' = (V, E, \{\boldsymbol{r}^{uv}\}_{(u,v)\in E})$, i.e., the original graph without node attributes, but equipped with the learned edge embeddings to give a final representation $M(G) := M_2(G')$.

The two GNNs $M_1$ and $M_2$ are trained end-to-end on the task. Note that $M_1$ can be any GNN designed to produce whole-graph embeddings and can take single-dimensional edge attributes, whilst $M_2$ can be any GNN that can take edge embeddings as input.

**Integration with language models.** While STAGE can incorporate LLM embeddings for textual attributes, our experiments show STAGE-edge-graphs performs better on numerical and categorical data (Section 4). The approaches can be complementary - initialize node embeddings with

LLM embeddings for textual attributes and edge embeddings with STAGE-edge-graphs for non-textual attributes.

**Modelling pairwise relations.** $\boldsymbol{S}^{uv}$ is only computed for *edges* $(u, v)$, and so can only model pairwise relations between nodes connected by an edge. In some cases, such as bipartite graphs, we find it beneficial to add extra edges between nodes of the same type (see Section 4 for details). In general, higher-order relations could also be modelled similarly, albeit at increased complexity. We leave exploration of higher-order relations to future work.

## 3. Statistical Underpinnings of STAGE

This section explains how STAGE achieves domain transferability. The central result is to show that STAGE generates representations capable of measuring dependencies among node attributes in graphs. This means that STAGE can ignore "absolute" attribute values, while still generalizing through analogous statistical dependencies of the attributes.

Our first step (Section 3.1) connects measures of statistical dependencies with a novel graph regression task. Then, Section 3.2 shows that our STAGE-edge-graphs (Definition 2.1) can lead to a compact model for this regression, with a variant that is invariant to a class of shifts between train and test attribute domains. The following theoretical results are meant to provide insights and are restricted to domains with a fixed number of attributes to simplify the proofs, extending them to variable size spaces is left as future work. Detailed proofs are provided in Appendix B.

### 3.1. Statistical Dependence as Graph Regression

We begin by introducing the framework for building what we call *feature hypergraphs*. We will show that feature hypergraphs can sufficiently encapsulate the statistical dependencies between attributes, while only leveraging the relative orders rather than the numerical values of the attribute, enabling it to be invariant to order-preserving transformations (formally defined in Definition B.2) to achieve domain transferability. In the following, we assume one attribute space defined over a totally ordered set (e.g., $\mathbb{R}^d$ for $d \geq 1$, where the total order $\leq_\tau$ is well defined), since the invariances of unordered sets are a special case (as these do not need order-preserving transformations). Before we describe how feature hypergraphs are built, we start with the concept of order statistic, which captures the relative ordering of the attribute values.

**Order statistic** (David & Nagaraja, 2004). Let $\mathrm{x}_1, \mathrm{x}_2, \ldots, \mathrm{x}_m$ be a sequence of $m \geq 2$ random variables from some unknown distribution $F$ over a totally ordered set (e.g., a convex set $\mathbb{F} \subseteq \mathbb{R}$). Its *order statistics* are defined as the sorted values $\mathrm{x}_{(1)} \leq \mathrm{x}_{(2)} \leq \cdots \leq \mathrm{x}_{(m)}$, where $\mathrm{x}_{(k)}$ denotes the $k$-th smallest value in the $m$ samples.

Consider a domain with $m$ entities (e.g., products in an appliance store), where each entity is characterized by $d$ attributes. Specifically, an entity $u$ can be represented by a (row) vector of random attribute variables, $\mathbf{x}^u = [\mathrm{x}_1^u, \mathrm{x}_2^u, \ldots, \mathrm{x}_d^u]$, where $\mathrm{x}_i^u$ describes the $i$-th attribute of entity $u$ that takes on values from the $i$-th attribute space $\mathbb{F}_i \subseteq \mathbb{R}$. With these variables, we define the (random) matrix $\mathbf{X} := [(\mathbf{x}^1)^T, (\mathbf{x}^2)^T, \ldots, (\mathbf{x}^m)^T]^T$ of shape $m \times d$. Alternatively, we can view $\mathbf{X}$ column-wise, where each attribute $i$ corresponds to a (column) random vector $\mathbf{x}_i = [\mathrm{x}_i^1, \mathrm{x}_i^2, \ldots, \mathrm{x}_i^m]^T$. Next, we introduce the order statistic for these attributes: let $\mathbf{x}_{i(k)}$ denote the $k$-th order statistic of $\{\mathrm{x}_i^1, \ldots, \mathrm{x}_i^m\}$. For instance, $\mathbf{x}_{i(1)} = \min\{\mathrm{x}_i^1, \ldots, \mathrm{x}_i^m\}$.

Given an input graph $G = (V, E, \boldsymbol{X})$, we regard it as a sample from some unknown distribution over all attributed graphs with $m$ entities and $d$ attributes, where $\boldsymbol{X}$ is a random variable with $\boldsymbol{X} = [\boldsymbol{x}_1, \ldots, \boldsymbol{x}_d]$. Consider the edges in $E$ as samples of pairs of nodes that give rise to the multiset of attributes of the endpoint nodes, $\mathcal{E} = \{\{(\boldsymbol{x}^u, \boldsymbol{x}^v) \mid (u, v) \in E\}\}$. Together with the order statistics, we now define the attribute hypergraph as follows:

**Definition 3.1** (Attribute hypergraph $\mathcal{F}_\mathcal{E}$). Given a multiset of attributes of the endpoint nodes $\mathcal{E} = \{\{(\boldsymbol{x}^u, \boldsymbol{x}^v) \mid (u, v) \in E\}\}$ of $m$ entities with totally ordered attribute spaces, the feature hypergraph $\mathcal{F}_\mathcal{E}$ is defined as follows. First, we label the graph with $m$. Then,

- For each order statistic $\boldsymbol{x}_{i(k)}$ of attribute $i$ and order $k$ ($1 \leq k \leq m$), there are 2 nodes, namely $(i, k, 1)$ and $(i, k, 2)$. In total, there are exactly $2md$ nodes in $\mathcal{F}_\mathcal{E}$ (attribute values need not be unique). Nodes $(i, k, 1)$ and $(i, k, 2)$ store a single attribute to mark their order: $k$.

- Let $o_i(u)$ be the order of the attribute value $\boldsymbol{x}_i^u$, i.e., $\boldsymbol{x}_{i(o_i(u))} = \boldsymbol{x}_i^u$. For each pair of attributes of endpoint nodes $(\boldsymbol{x}^u, \boldsymbol{x}^v) \in \mathcal{E}$, there is a hyperedge $H_{uv}$ in $\mathcal{F}_\mathcal{E}$ defined as

$$
\begin{aligned}
H_{uv} := \{(1, o_1(u), 1), (1, o_1(v), 2), \\
(2, o_2(u), 1), (2, o_2(v), 2), \ldots, \\
(d, o_d(u), 1), (d, o_d(v), 2)\}. \quad (4)
\end{aligned}
$$

Our first observation is that the feature hypergraph in Definition 3.1 perfectly captures the order statistics of the set $\mathcal{E}$ but discards the actual values of the attributes.

We now consider statistical tests that measure dependencies of the attributes of endpoint nodes. As an example, consider that if $(\boldsymbol{x}^u, \boldsymbol{x}^v) \in \mathcal{E}$ are samples (not necessarily independently sampled) from a bivariate distribution $(\mathbf{x}, \mathbf{x}') \sim F$, one may be interested in testing the hypothesis

$$
H_0 : F(\mathbf{x}, \mathbf{x}') = F_1(\mathbf{x})F_2(\mathbf{x}'),
$$

i.e., that $\mathbf{x}$ and $\mathbf{x}'$ are independent. Bell (1964); Berk & Bickel (1968) showed that over totally ordered sets, measures (e.g., $p$-values) of such hypothesis tests for pairwise independence ($H_0$ above) and higher-order conditional independence between multiple variables, have invariances that simplify the data representation to such a degree that the original values are discarded, retaining only the order relationships between the variable values. Any such test is therefore a rank test, i.e., it relies only on indices of the order statistic, not on the numerical values of the attributes.

Our first theoretical contribution is the observation that any statistical test that focuses on measuring the (conditional) dependencies of attributes of endpoint nodes in $\mathcal{E}$ can be defined as a graph regression task over the feature hypergraph $\mathcal{F}_{\mathcal{E}}$ of Definition 3.1.

**Theorem 3.2.** *Given a multiset of attributes of the endpoint nodes $\mathcal{E}$, the corresponding feature hypergraph $\mathcal{F}_{\mathcal{E}}$ (Definition 3.1) and a most-expressive hypergraph GNN encoder $M_{\theta^*}(\mathcal{F}_{\mathcal{E}})$, then any test $T(\mathcal{E})$ that focuses on measuring the dependence of the attributes of the endpoint nodes of $\mathcal{E}$ has an equivalent function $h$ within the space of Multilayer Perceptrons (MLPs) that depends solely on the graph representation $M_{\theta^*}(\mathcal{F}_{\mathcal{E}})$, i.e., $\exists h \in MLPs$ s.t. $T(\mathcal{E}) = h(M_{\theta^*}(\mathcal{F}_{\mathcal{E}}))$.*

Next we show that the hypergraph $\mathcal{F}_{\mathcal{E}}$ can be simplified with STAGE-edge-graph and that the ability to compute dependency measures can be made invariant to certain domain shifts between train and test.

### 3.2. Transferability of STAGE

The feature hypergraph $\mathcal{F}_{\mathcal{E}}$ in Definition 3.1 is used to obtain a maximal invariant graph representation via hypergraph GNN. This solution has a high computational cost from the use of hypergraph GNNs. Fortunately, we show that by assigning unique attribute identifiers to label the nodes of our STAGE-edge-graphs $\boldsymbol{G}(\boldsymbol{S}^{uv})$ (Definition 2.1), STAGE-edge-graphs are as informative as the corresponding feature hypergraphs, preserve the same invariances, while allowing the usage of (non-hypergraph) GNN encoders.

**Theorem 3.3.** *Given the attributes of the endpoint nodes $\mathcal{E}$ (Definition 3.1) of a graph $G = (V, E, \boldsymbol{X})$, there exists an optimal parameterization $\theta_g^*, \theta_s^*$ for a most expressive GNN encoder $M^g$ and a most-expressive multiset encoder $M^s$, respectively, such that $M_{\theta_s^*,\theta_g^*}(G) \coloneqq M_{\theta_s^*}^s(\{\{M_{\theta_g^*}^g(\boldsymbol{G}(\boldsymbol{S}^{uv})) : (u,v) \in E\}\})$ such that any test $T(\mathcal{E})$ that measures the dependence of $\mathcal{E}$'s attributes of the endpoint nodes has an equivalent function $h$ within the space of Multilayer Perceptrons (MLPs) that depends solely on the graph representation $M_{\theta_s^*,\theta_g^*}(G)$, i.e., $\exists h \in MLPs$ s.t. $T(\mathcal{E}) = h(M_{\theta_s^*,\theta_g^*}(G))$.*

Theorem 3.3 motivates the design of STAGE, which lever-

ages a GNN on STAGE-edge-graphs to obtain edge-level embeddings. However, the use of unique attribute identifiers in the STAGE-edge-graphs disrupts invariance to permutations in attribute order (e.g., U.S. shoe size appearing as the first attribute in one dataset and U.K. shoe size as the last attribute in another), thereby limiting its domain transferability. More broadly, we now describe all the invariances we want for STAGE to have in order to be robust to a class of attribute domain shifts.

**COGG invariances.** STAGE-edge-graphs facilitate domain transfer to distinct attribute domains. Intuitively, the full set of invariances required for domain transferability over $G = (V, E, \boldsymbol{X})$ consists of: (1) invariance or equivariance to transformations of attribute values that preserve the order statistic, (2) invariance or equivariance to permutations of attribute orders (columns of $\boldsymbol{X}$), and (3) invariance or equivariance to permutations of nodes in the graph, affecting $V$ (and consequently $E$) and the rows of $\boldsymbol{X}$. These invariances are formalized in Definition B.5 in Appendix B.4 through the actions of *component-wise order-preserving groupoid for graphs* (COGG). Importantly, groups are insufficient to capture these invariances because they assume transformations act within a single attribute domain. However, we are interested in transformations across distinct attribute spaces. Groupoids generalize groups by allowing these transformations between different domains, making them the natural choice for modeling the required invariances.

We now introduce our final theoretical contribution which establishes that STAGE achieves invariance to COGGs by design. This result shows that STAGE can provably achieve the zero-shot transferability to the class of attribute domain shifts defined by COGGs-type transformations.

**Theorem 3.4.** *STAGE is invariant to COGGs (Definition B.5).*

The proof sketch is as follows. From Theorem 3.3, STAGE achieves invariance to changes in attribute values, including their types, names, and semantics. Then, by dropping the attribute identifiers in STAGE-edge-graphs, we sacrifice maximal expressivity but ensure that STAGE is invariant to permutations of the attribute order. Finally, since STAGE employs a second GNN on the original input graph, using the embeddings of the STAGE-edge-graphs, while omitting the original node attributes, STAGE achieves invariance to node permutations. Thus, the method is invariant to COGGs.

## 4. Experiments

We demonstrate the effectiveness of STAGE across multiple experimental settings, focusing on small to medium-sized datasets. While the computational complexity scales linearly with the graph size and quadratically with the number of attributes, training on these datasets introduces only mod-

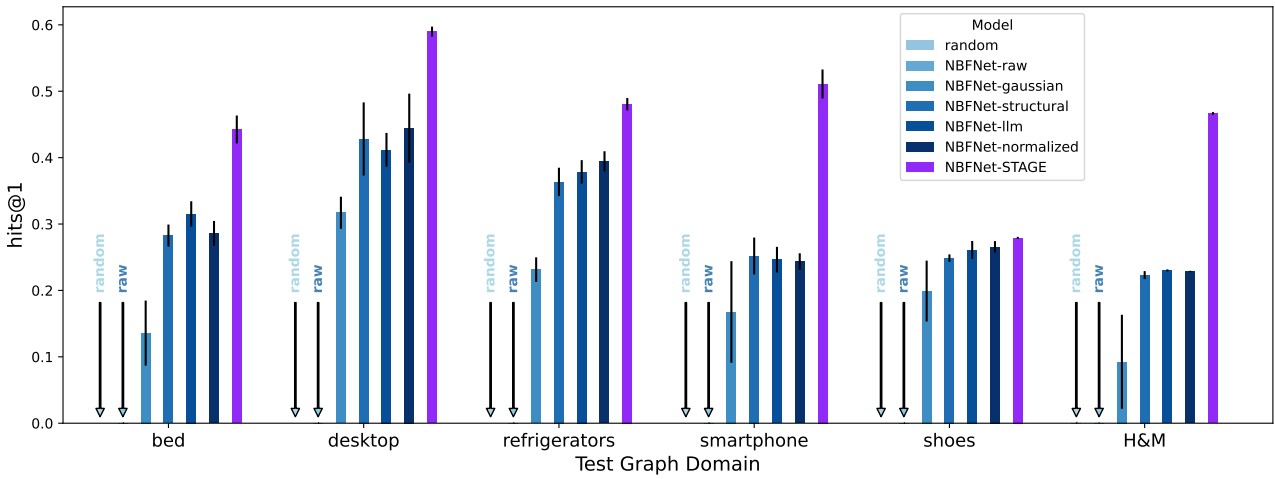

Figure 3: Zero-shot Hits@1 performance (higher is better) of STAGE and baselines, trained on four (or five) distinct E-Commerce Store domains and evaluated on the held-out domain (or H&M dataset). **NBFNet-STAGE consistently achieves the highest zero-shot accuracy across all test domains, with up to 103% improvement**.

erate computational overhead (e.g., 7.83% slower than the fastest baseline in link prediction; see Appendix H). Therefore, STAGE is highly effective in these settings, achieving strong generalization performance. In the following, we present our main results and refer to Appendix D for details. Our code is available at https://github.com/snap-stanford/stage-gnn/.

**Datasets.** To evaluate zero-shot generalization to graphs with new attributes, we consider datasets with distinct domain-specific attributes but a shared task. Our datasets contain graphs with up to 4k nodes, 50k edges, 16 attributes, representing small to medium-size real-world scenarios where STAGE is particularly effective. Due to space constraints, we introduce them below and refer to Appendix C.

*E-Commerce Stores dataset (link prediction).* We use data from a multi-category store (Kechinov, 2020) containing customer-product interactions (purchases, cart additions, views) over time. To simulate distinct single-category retailers, we partition the dataset into five domains, each representing a specialized store: *shoes*, *refrigerators*, *desktops*, *smartphones*, and *beds*. Each domain has its own customer base and product-specific attributes (e.g., *smartphones* have *display type*; *shoes* have *ankle height*). The task is to predict future customer-product interactions from past actions.

*H&M dataset (link prediction).* We use the H&M Personalized Fashion Recommendations dataset (Kaggle, 2021), which contains transactions from a large fashion retailer, to evaluate the zero-shot performance of models trained on E-Commerce Stores. All attributes, except for "price", differ from those in E-Commerce Stores. The task remains to predict customer-product interactions from past actions.

*Social network datasets (node classification): Friendster and Pokec.* We evaluate STAGE on two online social networks from different regions and user bases: Friendster (Teixeira et al., 2019) and Pokec (SNAP, 2012). Friendster nodes have attributes such as *age*, *gender*, *interests*, while Pokec nodes have *public profile status*, *completion percentage*, *region*, *age*, and *gender*. The task is to predict a node attribute common to both social networks using network structure and remaining node attributes. Since only *age* and *gender* are shared, we create two tasks: mask and predict *gender* (presented in this section), and mask and regress on *age* (discussed in Appendix E).

**Baselines.** We compare STAGE to several baselines designed to handle new node attributes. (1) *raw*: Projects each raw node attribute into a fixed-dimensional space with a linear transformation, before summing across the projected dimensions. (2) *gaussian*: Use Gaussian noise as node attributes (Sato et al., 2021; Abboud et al., 2021). (3) *structural*: Ignores node attributes entirely, using only the graph structure. (4) *llm*: Converts node attributes into textual descriptions and obtains embeddings using a pretrained encoder-only language model, taking only the node attributes as input (without graph structure) due to prompt length limitations, similar to PRODIGY (Huang et al., 2023). (5) *normalized*: Retains only continuous attributes and standardize them. For a fair comparison, all methods utilize the same underlying GNN architecture, NBFNet (Zhu et al., 2021c) for link prediction and GINE (Hu et al., 2020) for node classification. In Appendix F, we report additional experiments with other architectures. In addition to these baselines, we evaluate our approach against *GraphAny* (Zhao et al., 2024b), a recent method for domain transferability in node

classification tasks, but not applicable to link prediction.

## 4.1. Zero-Shot Link Prediction on Unseen Domains

We evaluate the performance of all methods on zero-shot generalization on the E-Commerce Stores dataset, training on four categories, and testing on the held-out fifth category.

**Results.** As shown in Figure 3, STAGE consistently outperforms all baselines in zero-shot Hits@1 across all test domains. Notable improvements include: 103% gain when testing on the *smartphone* category (0.51 vs 0.25 Hits@1), 40% on *bed* (0.44 vs 0.31), and 33% on *desktop* (0.59 vs 0.44) compared to the strongest baselines.

In Table 1, we report the average performance of each model, calculated by taking the results in which each domain is held out once and averaging the scores. Our evaluation also includes popular non-parametric link prediction approaches such as Common Neighbors, Adamic Adar, and Personalized PageRank, with results showing that STAGE substantially outperforms classical heuristic methods by 54%, 51%, and 3837% respectively on Hits@1, while maintaining similar performance advantages on MRR. Overall, STAGE achieves 41% higher average Hits@1 (0.46 vs 0.33) and 29% higher MRR (0.50 vs 0.38) against the strongest baseline (normalized), with lower variance across seeds. This emphasizes the benefit of STAGE in transforming node attributes into a *unified input space* using learned edge embedding via *STAGE-edge-graph*, including its stronger attribute representation capabilities than LLM-based encoding approaches in the medium-sized graphs considered in this work.

## 4.2. Cross-Dataset Zero-Shot Link Prediction

We evaluate models trained on E-Commerce Stores for zero-shot prediction on the H&M dataset, which has distinct customers, products, activity patterns and attributes.

Table 1 shows that the performance on H&M of STAGE when trained on E-Commerce Stores is virtually identical to its performance on the held-out category in E-Commerce Stores (0.46 vs. 0.46 Hits@1). This highlights the robustness of STAGE to domain shifts, as it maintains similar performance when transitioning from E-Commerce Stores, which primarily feature household items, electronics, and shoes, to H&M, which focuses on clothing with minimal overlap in product types.

In Hits@1, STAGE achieves a relative improvement of 103% over the best parametric baseline (*llm*) (0.46 vs. 0.23). Moreover, STAGE obtains a relative improvement of 202% against a supervised structural method trained and tested on H&M (*structural-supervised*). In MRR, STAGE achieves the highest score, outperforming the best baseline by 99%.

Moreover, STAGE demonstrates a substantial improvement

of 99% in Hits@1 over Adamic Adar (0.466 vs 0.2349), which performs the best among traditional heuristic methods on the H&M dataset. Similarly, STAGE outperforms Adamic Adar by 48% in MRR (0.4703 vs. 0.3184), further confirming the its superiority over classical link prediction heuristics in zero-shot scenarios.

## 4.3. Zero-Shot Node Classification on Unseen Domains

To validate our approach beyond link prediction and E-Commerce scenarios, we benchmark on a node classification task using two social network datasets, where the goal is to predict user *gender*. We train models on Friendster and evaluate zero-shot on Pokec.

Table 2 shows that STAGE achieves a 10.3% improvement over the best baseline (and lower variance), also surpassing the task-specific model GraphAny (Zhao et al., 2024b) and the cross-domain pretraining method GCOPE (Zhao et al., 2024a). This indicates that STAGE effectively captures attribute dependencies also in node classification tasks and outperforms all approaches by leveraging its unified input space obtained by the usage of the STAGE-edge-graphs.

## 4.4. Generalization When Training on Multiple Domains

We examine how the model performance varies with the number of training domains in E-Commerce Stores.

As shown in Figure 4, STAGE obtains improving zero-shot performance (both Hits@1 and MRR) with more training domains. While not the only method showing improvement, STAGE exhibits notably tighter interquartile ranges compared to the only other method exhibiting better performance with increasing domain (*gaussian*) at higher domain counts. Additionally, STAGE's lower whiskers consistently rise with more domains, showing also that its worst-case scenarios improve with more training data.

These results further validate that STAGE is capable of learning transferable patterns across domains through its defined *unified input space*. The consistent performance gains with additional training domains suggest that *STAGE-edge-graph* effectively captures generalizable dependencies between attributes, with more training domains enabling the learning of a broader range of dependencies. In contrast, baseline approaches that ignore attributes or use generic embeddings fail to leverage the additional training domains for improved cross-domain generalization.

## 5. Related Work

In this section, we present the most closely related works to our STAGE. A more in-depth comparison, along with additional related work, can be found in Appendix I.

Table 1: **NBFNet-STAGE outperforms all baselines in zero-shot Hits@1 and MRR (including supervised approaches) across the E-Commerce Stores and H&M datasets.** For the E-Commerce Stores, results are averaged across models trained on all combinations of four graph domains and tested on the remaining domain. For zero-shot test on H&M, models are trained on the five E-Commerce Stores domains. % **gain** shows relative improvement of STAGE over each baseline.

| Training: E-Commerce Stores | Test: Held-out E-Comm. Store | | | | Test: H&M Dataset | | | |
|---|---|---|---|---|---|---|---|---|
| Model | Hits@1 (↑) | % gain | MRR | % gain | Hits@1 (↑) | % gain | MRR (↑) | % gain |
| random | $0.0026 \pm 0.0000$ | 17615% | - | - | $0.0006 \pm 0.0000$ | 77667% | - | - |
| Common Neighbors | $0.2991 \pm 0.0006$ | 54% | $0.3942 \pm 0.0014$ | 26% | $0.2354 \pm 0.0000$ | 98% | $0.3179 \pm 0.0000$ | 48% |
| Adamic Adar | $0.3052 \pm 0.0007$ | 51% | $0.4001 \pm 0.0015$ | 24% | $0.2349 \pm 0.0000$ | 99% | $0.3184 \pm 0.0000$ | 48% |
| Personalized PageRank | $0.0117 \pm 0.0000$ | 3837% | $0.0714 \pm 0.0001$ | 596% | $0.0105 \pm 0.0000$ | 4344% | $0.0717 \pm 0.0000$ | 556% |
| NBFNet-raw | $0.0000 \pm 0.0000$ | $\infty$ | $0.0032 \pm 0.0009$ | 15434% | $0.0005 \pm 0.0004$ | 93220% | $0.0059 \pm 0.0011$ | 7871% |
| NBFNet-gaussian | $0.2101 \pm 0.0428$ | 119% | $0.2617 \pm 0.0459$ | 90% | $0.0925 \pm 0.0708$ | 404% | $0.1176 \pm 0.0756$ | 300% |
| NBFNet-structural | $0.3149 \pm 0.0253$ | 46% | $0.3721 \pm 0.0219$ | 34% | $0.2231 \pm 0.0060$ | 109% | $0.2302 \pm 0.0080$ | 104% |
| NBFNet-llm | $0.3226 \pm 0.0190$ | 43% | $0.3830 \pm 0.0145$ | 30% | $0.2302 \pm 0.0015$ | 103% | $0.2365 \pm 0.0021$ | 99% |
| NBFNet-normalized | $0.3269 \pm 0.0213$ | 41% | $0.3844 \pm 0.0159$ | 29% | $0.2286 \pm 0.0010$ | 104% | $0.2341 \pm 0.0018$ | 101% |
| NBFNet-structural-supervised | N/A | N/A | N/A | N/A | $0.1546 \pm 0.0084$ | 202% | $0.2103 \pm 0.0164$ | 124% |
| **NBFNet-STAGE (Ours)** | $\mathbf{0.4606} \pm 0.0123$ | 0% | $\mathbf{0.4971} \pm 0.0073$ | 0% | $\mathbf{0.4666} \pm 0.0020$ | 0% | $\mathbf{0.4703} \pm 0.0029$ | 0% |

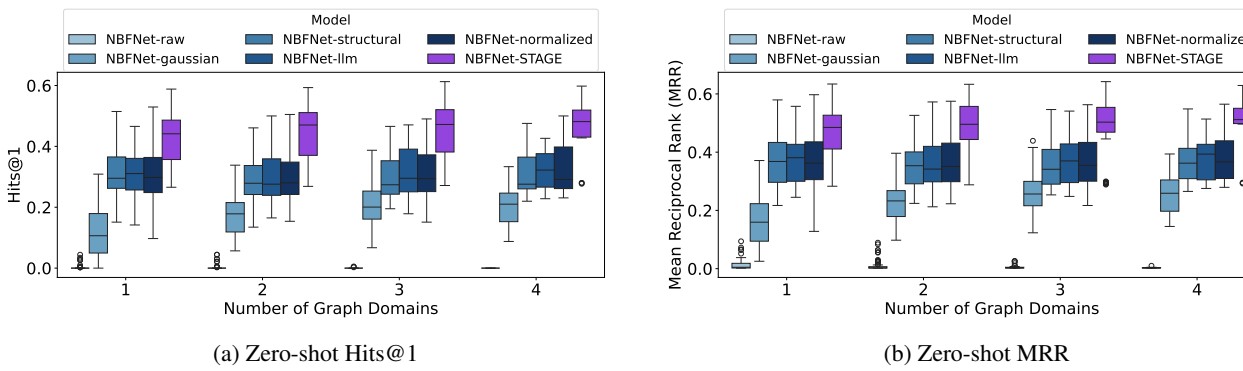

(a) Zero-shot Hits@1        (b) Zero-shot MRR

Figure 4: **The performance (both Hits@1 and MRR) of STAGE improves with more train domains, while this is not the case for other methods.** Box-plot distribution over all combinations of a fixed number of graph domains in the E-Commerce Stores dataset and testing on the held-out domain(s), averaged over random seeds.

**Graphs Generalization under Distribution Shifts.** Several works address distribution shifts between train and test graphs over the same attribute domain, such as Shen et al. (2023); Zhu et al. (2021b), which employ learned augmentations to mitigate the change in distribution in test. Meanwhile, extensive research has focused on domain adaptation for GNNs (Dai et al., 2022; Li et al., 2020; Kong et al., 2022; Pei et al., 2020; Veličković et al., 2019; Wiles et al., 2022; Zhang et al., 2019; Zhu et al., 2021a), which typically assume access to data in both source and target domains. In contrast, our work tackles the more challenging scenario of zero-shot generalization to unseen attribute domains. To the best of our knowledge, all out-of-distribution graph methods (Zhang et al., 2024a) do not address the attribute domain shifts we consider, which include changes in the number of attributes between train and test.

**Foundation Models for Graphs.** Developing foundation models for graph data is a growing research interest, aiming

to create versatile graph models capable of generalizing across different graphs and tasks (Mao et al., 2024). Initial efforts in this direction convert attributed graphs into texts and apply an LLM (Liu et al., 2024; Chen et al., 2024b;a; Tang et al., 2024; Zhao et al., 2023; He & Hooi, 2024; Huang et al., 2023). However, while promising, this methodology risks information loss and may limit transferability (Collins et al., 2024; Gruver et al., 2024; Schwartz et al., 2024). In contrast, non-LLM approaches attempt to directly address domain transferability in the attribute space (Xia & Huang, 2024; Lachi et al., 2024; Zhao et al., 2024b; Frasca et al., 2024; Yu et al., 2024; Zhao et al., 2024a), or by avoiding the use of node attributes entirely (Gao et al., 2023; Lee et al., 2023; Galkin et al., 2024; Zhang et al., 2024b). We provide details in Appendix I about why some of these approaches are not applicable as our baselines.

Table 2: Zero-shot test accuracy of STAGE and baselines on the Pokec dataset, trained on Friendster. % **gain** shows relative improvement of STAGE over each baseline.

| Model | Accuracy ($\uparrow$) | % gain |
|---|---|---|
| random | $0.500 \pm 0.0000$ | 30.4% |
| GINE-raw | $0.558 \pm 0.0829$ | 16.8% |
| GINE-gaussian | $0.588 \pm 0.0250$ | 10.9% |
| GINE-structural | $0.564 \pm 0.0466$ | 15.6% |
| GINE-llm | $0.550 \pm 0.0368$ | 18.5% |
| GINE-normalized | $0.541 \pm 0.0148$ | 20.5% |
| GraphAny | $0.591 \pm 0.0083$ | 10.3% |
| GCOPE | $0.535 \pm 0.0153$ | 21.9% |
| **GINE-STAGE (Ours)** | $\mathbf{0.652} \pm \mathbf{0.0042}$ | 0% |

## 6. Conclusion and Future Work

The challenge of learning universal graph representations that generalize across diverse attribute domains has limited progress in graph foundation models, mainly due to the lack of a unified input space to represent node attributes, which may vary in test graphs. In this paper, we proposed STAGE, which addresses this limitation by transforming diverse attribute spaces into a unified representation, learning statistical dependencies between attributes instead of relying on their absolute values. By demonstrating that these dependencies remain invariant under certain domain shifts, STAGE provides theoretical foundations for zero-shot generalization across graphs with differing attribute spaces. Our strong empirical results on medium-sized datasets demonstrate the practical effectiveness of this approach.

While STAGE represents a meaningful step forward, it also highlights opportunities for future research. The unified input space we introduce could serve as a basis for developing graph foundation models that can learn from diverse graph datasets at scale, reducing the quadratic complexity of STAGE. However, realizing this potential will require addressing additional challenges, such as developing architectures to capture complex high-order attribute dependencies and scaling to large graph collections.

## Acknowledgments

BR and BB acknowledge support from the National Science Foundation (NSF) awards CCF-1918483, CAREER IIS-1943364 and CNS-2212160, an Amazon Research Award, and AnalytiXIN, Wabash Heartland Innovation Network (WHIN), Ford, NVidia, CISCO, and Amazon. Computing infrastructure was supported in part by CNS-1925001 (CloudBank). This work was supported in part by AMD under the AMD HPC Fund program.

JL gratefully acknowledges the support of NSF under Nos. OAC-1835598 (CINES), CCF-1918940 (Expeditions), DMS-2327709 (IHBEM), IIS-2403318 (III); Stanford Data Applications Initiative, Wu Tsai Neurosciences Institute, Stanford Institute for Human-Centered AI, Chan Zuckerberg Initiative, Amazon, Genentech, GSK, Hitachi, SAP, and UCB. The content is solely the responsibility of the authors and does not necessarily represent the official views of the funding entities.

## Impact Statement

This paper presents work whose goal is to advance the field of Machine Learning. There are many potential societal consequences of our work, none which we feel must be specifically highlighted here.

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

# A. Pseudocode of STAGE Algorithm

In this section, we present the detailed pseudocode for STAGE's two main components: (1) the STAGE-edge-graphs construction algorithm (Algorithm 1) that captures statistical dependencies between attributes, and (2) the forward pass (Algorithm 2) that uses these STAGE-edge-graphs to generate the final graph representation. The STAGE-edge-graphs construction creates a complete graph for each edge in the input graph, where nodes represent attributes and edge weights capture conditional probabilities between attribute pairs. The algorithm handles both totally ordered and unordered attributes. The forward pass then processes these STAGE-edge-graphs using two GNNs - one to generate edge embeddings from the STAGE-edge-graphs, and another to produce the final graph representation using these embeddings.

---

**Algorithm 1** STAGE-edge-graphs Construction

---

 1: **Input:** Graph $G = (V, E, \boldsymbol{X})$ with node attributes $\boldsymbol{X} = \{\boldsymbol{x}^v\}_{v \in V}$
 2: **Output:** STAGE-edge-graphs $\{\boldsymbol{G}(\boldsymbol{S}^{uv})\}_{(u,v) \in E}$
 3: **for** each $(u, v) \in E$ **do**
 4:     Initialize $\boldsymbol{S}^{uv} \in \mathbb{R}^{2d \times 2d}$ with zeros
 5:     **for** $i = 1$ to $d$ **do**
 6:       **if** $\boldsymbol{x}_i^u$ is totally ordered **then**
 7:         $\boldsymbol{S}_{ii}^{uv} \leftarrow \mathbb{P}(\mathrm{x}_i^A \le \boldsymbol{x}_i^u)$
 8:       **else**
 9:         $\boldsymbol{S}_{ii}^{uv} \leftarrow \mathbb{P}(\mathrm{x}_i^A = \boldsymbol{x}_i^u)$
10:       **end if**
11:       **if** $\boldsymbol{x}_i^v$ is totally ordered **then**
12:         $\boldsymbol{S}_{(i+d)(i+d)}^{uv} \leftarrow \mathbb{P}(\mathrm{x}_i^B \le \boldsymbol{x}_i^v)$
13:       **else**
14:         $\boldsymbol{S}_{(i+d)(i+d)}^{uv} \leftarrow \mathbb{P}(\mathrm{x}_i^B = \boldsymbol{x}_i^v)$
15:       **end if**
16:     **end for**
17:     **for** $i = 1$ to $2d$ **do**
18:       **for** $j = 1$ to $2d, j \ne i$ **do**
19:         $(x_i, x_j) \leftarrow \text{GetAttributePair}(\boldsymbol{x}^u, \boldsymbol{x}^v, i, j, d)$
20:         **if** $x_i, x_j$ are totally ordered **then**
21:           $\boldsymbol{S}_{ij}^{uv} \leftarrow \mathbb{P}(\mathrm{x}_i \le x_i | \mathrm{x}_j \le x_j)$
22:         **else if** $x_i$ unordered, $x_j$ totally ordered **then**
23:           $\boldsymbol{S}_{ij}^{uv} \leftarrow \mathbb{P}(\mathrm{x}_i = x_i | \mathrm{x}_j \le x_j)$
24:         **else if** $x_i$ totally ordered, $x_j$ unordered **then**
25:           $\boldsymbol{S}_{ij}^{uv} \leftarrow \mathbb{P}(\mathrm{x}_i \le x_i | \mathrm{x}_j = x_j)$
26:         **else**
27:           $\boldsymbol{S}_{ij}^{uv} \leftarrow \mathbb{P}(\mathrm{x}_i = x_i | \mathrm{x}_j = x_j)$
28:         **end if**
29:       **end for**
30:     **end for**
31:     $\boldsymbol{G}(\boldsymbol{S}^{uv}) \leftarrow \text{CreateCompleteGraph}(2d)$
32:     **for** $i = 1$ to $2d$ **do**
33:       $\text{SetNodeAttribute}(\boldsymbol{G}(\boldsymbol{S}^{uv}), i, \boldsymbol{S}_{ii}^{uv})$
34:     **end for**
35:     **for** $i = 1$ to $2d$ **do**
36:       **for** $j = 1$ to $2d, j \ne i$ **do**
37:         $\text{SetEdgeAttribute}(\boldsymbol{G}(\boldsymbol{S}^{uv}), (i, j), \boldsymbol{S}_{ij}^{uv})$
38:       **end for**
39:     **end for**
40: **end for**

---

---

**Algorithm 2** STAGE Forward Pass

---

1: **Input:** Graph $G = (V, E, \boldsymbol{X})$, GNNs $M_1$ and $M_2$
2: **Output:** Graph representation $M(G)$
3: $\{\boldsymbol{G}(\boldsymbol{S}^{uv})\}_{(u,v) \in E} \leftarrow$ ConstructEdgeGraphs($G$)
4: **for** each $(u, v) \in E$ **do**
5: $\quad \boldsymbol{r}^{uv} \leftarrow M_1(\boldsymbol{G}(\boldsymbol{S}^{uv}))$
6: **end for**
7: $G' \leftarrow (V, E, \{\boldsymbol{r}^{uv}\}_{(u,v) \in E})$
8: $M(G) \leftarrow M_2(G')$
9: return $M(G)$

---

# B. Proofs and Additional Theoretical Results

## B.1. Groupoids

**Definition B.1** (Groupoids). A *groupoid* $\mathcal{G}$ consists of the following elements:

1. A collection of distinct *spaces*, denoted as Spaces($\mathcal{G}$).

2. A set of *transformations* (also called morphisms) between these spaces, denoted as Trans($\mathcal{G}$).

3. Each transformation $f \in$ Trans($\mathcal{G}$) maps one space in Spaces($\mathcal{G}$) to another space (or potentially to itself), denoted as $f : X \to Y$, where $X, Y \in$ Spaces($\mathcal{G}$).

4. There is a rule for combining transformations: for any two transformations $f : X \to Y$ and $g : Y \to Z$, their *composition* results in a transformation $g \circ f : X \to Z$.

5. Each space $S \in$ Spaces($\mathcal{G}$) has an *identity transformation* $\mathrm{id}_S : S \to S$ that maps $S$ to itself, such that for any space $X \in$ Spaces($\mathcal{G}$) and any transformation $f_1 : S \to X$ and $f_2 : X \to S$, it guarantees $f_1 \circ \mathrm{id}_S = f_1$ and $\mathrm{id}_S \circ f_2 = f_2$.

6. Every transformation $f : X \to Y$ has a unique *inverse transformation* $f^{-1} : Y \to X$ such that $f^{-1} \circ f = \mathrm{id}_X$ and $f \circ f^{-1} = \mathrm{id}_Y$.

## B.2. Statistical tests as graph regression on feature hypergraphs

To prove the result of Theorem 3.2, we will first show an intermediate result using the notion of *maximal invariants*. Let $\mathcal{G}$ be a transformation group acting on a space $\mathbb{X}$. A function $M : \mathbb{X} \to \mathbb{Z}$ is said to be maximal invariant if it is invariant to transformations of $\mathcal{G}$ and if $\forall x_1, x_2 \in \mathbb{X}$, $M(x_1) = M(x_2)$ implies $x_2 = g \circ x_1$ for some group action $g \in \mathcal{G}$, that is, if $M$ is constant on the orbits but for each orbit, it takes on a different value (Lehmann et al., 2005, pp. 214). A *maximal invariant* is a representation theory counterpart of *sufficient statistics*.

Our intermediate result will show that the feature hypergraph admits a graph representation that is a maximal invariant. But first, we need to formally define the class of invariances, which we show later is essential for STAGE's domain transferability. Since we are interested in attribute spaces of distinct domains, rather than using groups (which involve automorphisms mapping a space onto itself), we will use groupoids (Definition B.1). Groupoids generalize the concept of groups by allowing transformations between multiple spaces. In a group, all transformations map a space onto itself, while in a groupoid, transformations can map between different spaces, but must still be invertible.

**Definition B.2** (Component-wise order-preserving groupoids for attributes (COGF)). Let $\mathbb{X}_1, \mathbb{X}_2$ be two attribute spaces, both with $d$ attribute dimensions. An attribute transformation $f : \mathbb{X}_1 \to \mathbb{X}_2$ is said to be *component-wise order-preserving* if it can be decomposed into a set of maps $f_1, \ldots, f_d$, where each $f_i$ maps the $i$-th dimension of $\mathbb{X}_1$ to the $i$-th dimension of $\mathbb{X}_2$ and is a homomorphism that preserves the total order in $\mathbb{X}_1$, and all dimensions of both $\mathbb{X}_1$ and $\mathbb{X}_2$ have a mapping.

Given an attributes of the endpoint nodes $\mathcal{E} = \{\{(\boldsymbol{x}^u, \boldsymbol{x}^v) \mid (u, v) \in E\}\}$ and a groupoid action $f$ from the COGF (Definition B.2), we define how $f$ acts on $\mathcal{E}$ as follows:

$$f(\mathcal{E}) = \{\{(f(\boldsymbol{x}^u), f(\boldsymbol{x}^v)) \mid (u, v) \in E\}\}.$$

Now, we are ready to establish the intermediate result as follows.

**Lemma B.3.** *Given a multiset of attributes of the endpoint nodes $\mathcal{E}$ and the feature hypergraph $\mathcal{F}_{\mathcal{E}}$ (Definition 3.1). There exists a parameterization $\theta^*$ for a maximally expressive hypergraph GNN encoder $M$ such that $M_{\theta^*}(\mathcal{F}_{\mathcal{E}})$ is a maximal invariant under COGFs (Definition B.2).*

*Proof.* Let $\mathcal{V}(\mathcal{F}_{\mathcal{E}})$ be set of labeled nodes (labeled with the feature id and the order statistic position) of $\mathcal{F}_{\mathcal{E}}$, let $\mathcal{H}(\mathcal{F}_{\mathcal{E}})$ be set of hyperedges of $\mathcal{F}_{\mathcal{E}}$, and let $m(\mathcal{F}_{\mathcal{E}})$ be the number of entities from $\mathcal{E}$, which are labeled with the entire graph during creation. Given two hypergraphs $\mathcal{F}_{\mathcal{E}_1}, \mathcal{F}_{\mathcal{E}_2}$, we define $\mathcal{F}_{\mathcal{E}_1} = \mathcal{F}_{\mathcal{E}_2}$ if and only if $\mathcal{V}(\mathcal{F}_{\mathcal{E}_1}) = \mathcal{V}(\mathcal{F}_{\mathcal{E}_2})$, $\mathcal{H}(\mathcal{F}_{\mathcal{E}_1}) = \mathcal{H}(\mathcal{F}_{\mathcal{E}_2})$, and $m(\mathcal{F}_{\mathcal{E}}) = m(\mathcal{F}_{\mathcal{E}'})$. Note that since the node in the feature hypergraph are always labeled, a most expressive hypergraph GNN $M_{\theta^*}$ will ensure that $M_{\theta^*}(\mathcal{F}_{\mathcal{E}_1}) = M_{\theta^*}(\mathcal{F}_{\mathcal{E}_2})$ if and only if $\mathcal{F}_{\mathcal{E}_1} = \mathcal{F}_{\mathcal{E}_2}$.

Let $\mathcal{G}$ be the COGF (Definition B.2) and let $f \in \mathcal{G}$ be an arbitrary groupoid action of COFG. To show invariance, the goal is to show that $M_{\theta^*}(\mathcal{F}_{\mathcal{E}}) = M_{\theta^*}(\mathcal{F}_{f(\mathcal{E})})$ for any $\mathcal{E}$. Because $M_{\theta^*}$ is most expressive, this is equivalent to showing $\mathcal{F}_{\mathcal{E}} = \mathcal{F}_{f(\mathcal{E})}$.

Let $\mathcal{V}(\mathcal{F}_{\mathcal{E}}) = \{(i, k, l)\}_{i \in [d], k \in [m], l \in \{1,2\}}$. We first observe that since $f$ acts on individual feature values, it does not change the total number of entities. Hence, the set of hypergraph nodes remain unchanged, $\mathcal{V}(\mathcal{F}_{f(\mathcal{E})}) = \mathcal{V}(\mathcal{F}_{\mathcal{E}}) = m$.

For the edges, consider an arbitrary hyperedge $H_{uv}$ in $\mathcal{F}_{\mathcal{E}}$. Then, because $f$ is a COGF, it preserves the order statistics of all feature values. Thus, the order of the feature value $o_i(u)$ from $\mathcal{E}$ remains the the same as $o'_i(u)$ from $f(\mathcal{E})$, for all $i$ and $u$. Hence, $H_{uv}$ is also a hyperedge in $\mathcal{F}_{f(\mathcal{E})}$. Similarly, because $f$ has an inverse $f^{-1}$, we can show that for every edge $H'_{uv}$ in $\mathcal{F}_{f(\mathcal{E})}$, it is also in $\mathcal{F}_{\mathcal{E}}$ under the transformation $f^{-1}$. Thus, $\mathcal{H}(\mathcal{F}_{\mathcal{E}}) = \mathcal{H}(\mathcal{F}_{f(\mathcal{E})})$ and so $\mathcal{F}_{\mathcal{E}} = \mathcal{F}_{f(\mathcal{E})}$, and therefore $M_{\theta}(\mathcal{F}_{\mathcal{E}}) = M_{\theta}(\mathcal{F}_{f(\mathcal{E})})$.

To show maximality, let $M_{\theta^*}(\mathcal{F}_{\mathcal{E}}) = M_{\theta}(\mathcal{F}_{\mathcal{E}'})$ for some $\mathcal{E}$ and $\mathcal{E}'$. Our goal is to show that $\mathcal{E}$ and $\mathcal{E}'$ are on the same orbit, i.e. there exists a $f \in \mathcal{G}$ such that $f(\mathcal{E}) = \mathcal{E}'$.

Because $M_{\theta^*}$ is most expressive, we know $\mathcal{F}_{\mathcal{E}} = \mathcal{F}_{\mathcal{E}'}$. This implies that $\mathcal{V}(\mathcal{F}_{\mathcal{E}}) = \mathcal{V}(\mathcal{F}_{\mathcal{E}'})$ and $|\mathcal{H}(\mathcal{F}_{\mathcal{E}})| = |\mathcal{H}(\mathcal{F}_{\mathcal{E}'})|$. First, Let $m = |\mathcal{V}(\mathcal{F}_{\mathcal{E}})| = |\mathcal{V}(\mathcal{F}_{\mathcal{E}'})|$. And since $\mathcal{V}(\mathcal{F}_{\mathcal{E}}) = \mathcal{V}(\mathcal{F}_{\mathcal{E}'})$, we also know both $\mathcal{E}$ and $\mathcal{E}'$ must have the same number of features. Denote it $d$. In addition, because $\mathcal{H}(\mathcal{F}_{\mathcal{E}}) = \mathcal{H}(\mathcal{F}_{G'})$, we have $|E| = |E'|$. Second, pick any endpoint features $(\boldsymbol{x}^u, \boldsymbol{x}^v) \in \mathcal{E}$, and let $H_{uv} \in \mathcal{H}(\mathcal{F}_{\mathcal{E}})$ be the corresponding hyperedge. We know that $H_{uv} \in \mathcal{H}(\mathcal{F}_{\mathcal{E}'})$ as well. Hence, there exists an counterpart endpoint features $(\boldsymbol{x}^{u'}, \boldsymbol{x}^{v'}) \in \mathcal{E}'$ such that

$$\forall 1 \leq i \leq d, o_i(u) = o'_i(u') \text{ and } o_i(v) = o'_i(v'),$$

where $o_i(\cdot)$ is the order of values of $i$-th feature in $\mathcal{F}_{\mathcal{E}}$ and $o'_i(\cdot)$ the order of values of $i$-th feature in $\mathcal{F}_{\mathcal{E}'}$. Thus, we can construct a COGF groupoid action $f$ as follows:

Let $f$ be decomposed into a set of maps $f_1, \ldots, f_d$ for every feature dimension $i$. Each $f_i$ is a piecewise linear function $f_i$ defined as follows:

$$f_i(a) = \begin{cases} a - (\boldsymbol{x})_{i(0)} + (\boldsymbol{x}')_{i(0)} & \text{if } a < (\boldsymbol{x})_{i(0)} \\ (\boldsymbol{x}')_{i(k)} & \text{if } a = \boldsymbol{x}_{i(k)} \text{ for some } k : 1 \leq k \leq m'_i \\ (\boldsymbol{x}')_{i(k_0)} + \frac{(\boldsymbol{x}')_{i(k_1)} - (\boldsymbol{x}')_{i(k_0)}}{(\boldsymbol{x})_{i(k_1)} - (\boldsymbol{x})_{i(k_0)}}(a - (\boldsymbol{x})_{i(k_0)}) & \text{if } \boldsymbol{x}_{i(k_0)} < a < \boldsymbol{x}_{i(k_1)} \text{ for some} \\ & \qquad k_0, k_1 : 1 \leq k_0 < k_1 \leq m'_i \\ a - (\boldsymbol{x})_{i(m'_i)} + (\boldsymbol{x}')_{i(m'_i)} & \text{if } a > (\boldsymbol{x})_{i(m'_i)} \end{cases}$$

Since each $f_i$ is a piecewise linear function that strictly increases, each of them preserves the order of feature values. And since $f$ can be decomposed into $f_i$'s, $f$ is a COGF groupoid action.

Hence, we have showed that there exists a $f$ such that $f(\mathcal{E}) = \mathcal{E}'$ which shows maximality. Hence completing the proof.

$\square$

Based on Lemma B.3, we are ready to prove that measuring dependencies of the features $(\boldsymbol{x}^u, \boldsymbol{x}^v) \in \mathcal{E}$ under COGF invariances can be defined as depending only on a most-expressive GNN encoding of the feature hypergraph $\mathcal{F}_{\mathcal{E}}$. In short, this is because any hypothesis test $T(\mathcal{E})$ that can be expressed as rank test is invariant to COGF, and any invariant function can necessarily be expressed as depending only on a maximal invariant.

**Theorem 3.2.** *Given a multiset of attributes of the endpoint nodes $\mathcal{E}$, the corresponding feature hypergraph $\mathcal{F}_{\mathcal{E}}$ (Definition 3.1) and a most-expressive hypergraph GNN encoder $M_{\theta^*}(\mathcal{F}_{\mathcal{E}})$, then any test $T(\mathcal{E})$ that focuses on measuring the dependence of the attributes of the endpoint nodes of $\mathcal{E}$ has an equivalent function $h$ within the space of Multilayer Perceptrons (MLPs) that depends solely on the graph representation $M_{\theta^*}(\mathcal{F}_{\mathcal{E}})$, i.e., $\exists h \in$ MLPs s.t. $T(\mathcal{E}) = h(M_{\theta^*}(\mathcal{F}_{\mathcal{E}}))$.*

*Proof.* We first note that any test $T(\mathcal{E})$ that focuses on measuring the dependence or independence of endpoint features of $\mathcal{E}$ is necessarily a rank test that relies solely on the indices of the order statistics rather than the numerical values of the features (Bell, 1964; Berk & Bickel, 1968). As such, $T(\mathcal{E})$ is invariant to COGFs (Definition B.2). Now, we show that given $\mathcal{E}$, $\mathcal{F}_{\mathcal{E}}$, and a most expressive hypergraph GNN encoder $M_{\theta^*}$, there exists an $h$ such that $T(\mathcal{E}) = h(M_{\theta^*}(\mathcal{F}_{\mathcal{E}}))$.

For any $\mathcal{E}_1, \mathcal{E}_2$, we know that if $M_{\theta^*}(\mathcal{F}_{\mathcal{E}_1}) = M_{\theta^*}(\mathcal{F}_{\mathcal{E}_2})$, then $f(\mathcal{E}_1) = \mathcal{E}_2$ for some groupoid action $f$ in COGF (Lemma B.3). Then, because $T$ is invariant to $f$, we have that $T(\mathcal{E}_1) = T(f(\mathcal{E}_1)) = T(\mathcal{E}_2)$. Hence, each value of $M_{\theta^*}(\mathcal{F}_{\mathcal{E}})$ is associated with no more than one value of $T(\mathcal{E})$. In other words, there exists a mapping $h^*$ such that $h^*(M_{\theta^*}(\mathcal{F}_{\mathcal{E}})) = T(\mathcal{E})$.

Since MLPs are universal function approximators (Leshno et al., 1993), there exists a MLP $h$ that approximates $h^*$, i.e., $h(M_{\theta^*}(\mathcal{F}_{\mathcal{E}})) = T(\mathcal{E})$. $\qquad\square$

### B.3. Correspondence between STAGE-edge-graphs and feature hypergraphs

For the proof of Theorem 3.3, we first prove an intermediate result, which establishes that there exists a bijective mapping between the feature hypergraph $\mathcal{F}_{\mathcal{E}}$ and the multiset of stage graphs, $\mathbb{S}_E := \{\{G(S^{uv}) \mid (u,v) \in E\}\}$, where each STAGE-edge-graph is equipped with unique feature ids. In the case of repeated feature values, we will show a bijective mapping to a collapsed feature hypergraph, where the nodes corresponding to the repeated feature values are collapsed into one single node, with its order $k$ being the smallest order of these repeated values. We denote by $n_i'$ the number of unique feature values of feature $i$. We note that such a collapsed feature hypergraph in the case of repeated feature values will provide a representation that is stabler than the traditional rank tests, as repeated values will translates into uncertainty or noise in the rank test results, whereas our feature hypergraph representation will remain stable.

**Lemma B.4.** *There exists a bijective mapping $\mathcal{I}$ between the multiset of STAGE-edge-graphs $\mathbb{S}_E :=$ $\{\{G(S^{uv}) \mid (u,v) \in E\}\}$ with unique feature ids and the feature hypergraph $\mathcal{F}_{\mathcal{E}}$ (Definition 3.1).*

*Proof.* Let $G = (V, E, X)$ be an input graph and let $\mathcal{E} = \{\{(x^u, x^v) \mid (u,v) \in E\}\}$ be the corresponding multiset of attributes of the endpoint nodes. We assume that each stage graph $G(S^{uv}) \in \mathbb{S}_E$ has nodes labeled as follows: the node associated with $i$-th feature for the source node $u$ is labeled with $(i, 1)$, and the node associated with $i$-th feature for the target node $v$ is labeled with $(i, 2)$, for every feature $i \in [d]$. Thus, given the graph $G(S^{uv})$, we can recover weighted adjacency matrix $S^{uv}$, and so there is a one-to-one mapping between them. Hence, for the following discussion, we refer to $G(S^{uv})$ and $S^{uv}$ interchangeably.

We first show that, given $\mathbb{S}_E$, we can construct $\mathcal{F}_{\mathcal{E}}$.

**Construct $\mathcal{I} : \mathcal{I}(\mathbb{S}_E) = \mathcal{F}_{\mathcal{E}}$:**

We first construct the set of feature hypergraph nodes. For every feature $i$, collect the multiset $Q_{i1} = \{\{S_{ii}^{uv}\}\}_{(u,v) \in E}$ and $Q_{i2} = \left\{\left\{S_{(i+d)(i+d)}^{uv}\right\}\right\}_{(u,v) \in E}$ and let $Q_i = Q_{i1} \cup Q_{i2}$. In words, $Q_{i1}$ collects the i-th feature's empirical c.d.f., $S_{ii}^{uv} = p(x_i^u) = \mathbb{P}(x_i \leq x_i^u)$, of the source node $u$ of all edges. Similarly, $Q_{i2}$ collects the i-th feature's empirical c.d.f., $S_{(i+d)(i+d)}^{uv} = p(x_i^v) = \mathbb{P}(x_i \leq x_i^v)$, of the target node $v$ of all edges. Note that $Q_i$ is a multiset, so if there are multiple nodes $u$ (or $v$) with the same i-th feature value $x_i^u$ (or $x_i^v$), they will have the same empirical c.d.f. $p(x_i^u)$ (or $p(x_i^v)$), and thus $Q_i$ will record the multiplicity (number of occurrence) of such repeated c.d.f. values.

Sort the unique values in the multiset $Q_i$ in ascending order and denote the sorted sequence of unique values as $S_i = (s_1, s_2, \ldots, s_{m_i'})$ where $s_l \in Q_i$ for each $l \in [m_i']$ where $m_i' \leq m$ is the total number of unique values for feature $i$ (if all values have multiplicity of 1, then $m_i' = m$). Denote $n_i(s_l)$ the multiplicity of the value $s_l$ in the multiset $Q_i$. Then, we can recover the feature hypergraph's nodes corresponding to the $i$-th feature as follows:

- For the smallest feature value, Ccnstruct the two nodes labeled $(i, 1, 1)$ and $(i, 1, 2)$.

- For $l \in \{2, 3, \ldots, m_i'\}$ and $s_l \in S_i$, construct the two nodes labeled as $(i, l-1+n_i(s_{l-1}), 1)$ and $(i, l-1+n_i(s_{l-1}), 2)$. $l-1+n_i(s_{l-1})$ is the order of the feature value $s_l$, accounting for multiplicity.

Repeating the above process for all features $i$ will recover the node set of the feature hypergraph.

Then, we reconstruct the multiset of hyperedges. Take any $\boldsymbol{S}^{uv} \in \mathbb{S}_E$. Again, for every feature $i$, we have $\boldsymbol{S}^{uv}_{ii} = p(x^u_i)$ denoting the empirical c.d.f. of the i-th feature of the source node $u$. Let $N_{iu} = \{(i, k_{i,1}, 1), \ldots, (i, k_{i,m'_i}, 1)\}$, where the $k_{i,l}$'s are the orders ($l \in [m'_i]$). $N_{iu}$ then is the subset of hypernodes for feature $i$ associated with the source node $u$ in the original edge in the input graph. Now, let $l'$ be the smallest integer in $[m'_i]$ such that $k_{i,l'} > \boldsymbol{S}^{uv}_{ii} = p(x^u_i)$, and let $k^u_i = k_{i,l'} - 1$. Then, $k^u_i$ is the order of the i-th feature of node $u$, i.e., $k^u_i = o_i(u)$.

Similarly, for every feature $i$, we have $\boldsymbol{S}^{uv}_{(i+d)(i+d)} = p(x^v_i)$, the empirical marginal c.d.f. when node $v$ is the target node of an edge. Let $N_{iv} = \{(i, k_{i,1}, 2), \ldots, (i, k_{i,m'_i}, 2)\}$. Let $l''$ be the smallest integer in $[m'_i]$ such that $k_{l''} > \boldsymbol{S}^{uv}_{(i+d)(i+d)} = p(x^v_i)$. Then, let $k^v_i = k_{l''} - 1$, and this is the order of the i-th feature of node $v$, i.e., $k^v_i = o_i(v)$.

Hence, we have recovered the hyperedge:

$$H_{uv} = \{(i, k^u_i, 1)\}_{i \in [d]} \cup \{(i, k^v_i, 2)\}_{i \in [d]}.$$

where $k^u_i$ and $k^v_i$ are defined as above.

Repeat the above process for every $\boldsymbol{S}^{uv} \in \mathbb{S}_E$, then we recover the entire multiset of hyperedges for the feature hypergraph.

**Construct $\mathcal{I}^{-1} : \mathcal{I}^{-1}(\mathcal{F}_\mathcal{E}) = \mathbb{S}_E$.**

Given a feature hypergraph $\mathcal{F}_\mathcal{E}$ with $\mathcal{V}(\mathcal{F}_\mathcal{E})$ the set of nodes and $\mathcal{H}(\mathcal{F}_\mathcal{E})$ the multiset of hyperedges. Our goal is to reconstruct the multiset of STAGE-edge-graphs $\mathbb{S}_E = \{\{\boldsymbol{G}(\boldsymbol{S}^{uv}) \mid (u, v) \in E\}\}$ for some underlying edge set $E$.

Pick any hyperedge $H = \{(i, k^1_i, 1)\}_{i \in [d]} \cup \{(i, k^2_i, 2)\}_{i \in [d]} \in \mathcal{H}(\mathcal{F}_\mathcal{E})$, where $k^1_i = o_i(u)$ is the order of i-th feature value for some unknown node $u$ and $k^2_i = o_i(v)$ the order of i-th feature value for some unknown node $v$. We first construct the corresponding STAGE-edge-graph adjacency matrix, which we denote $\boldsymbol{S}^H$. Once $\boldsymbol{S}^H$ is obtained, then we have the STAGE-edge-graph $\boldsymbol{G}(\boldsymbol{S}^H)$.

First, we construct the diagonal entries of $\boldsymbol{S}^H$ as follows. Note that the entire hypergraph is labeled with an integer $m$, which indicate the total number of entities (nodes) in the original input graph. Hence, we can recover the marginal empirical c.d.f. of the i-th feature value of each entity. Specifically, for every feature $i$, we have $k^1_i$ from the hyperedge $H$, denoting the order of i-th feature value of the underlying source node $u$ of an edge in the original input graph. If there is another hypergraph node $(i, k', 1) \in \mathcal{V}(\mathcal{F}_\mathcal{E})$ such that $k' > k^1_i$, then let $n^1_i = k' - 1$. Otherwise, let $n^1_i = m$. Thus, $n^1_i$ indicates the total number of nodes in the original input graph that have the i-th feature values smaller than or equal to the i-th feature value of the current node $u$. Note that $n^1_i$ accounts for multiplicity, if there were multiple nodes having the same i-th feature value as this node. Hence, let $\boldsymbol{S}^H_{ii} = n^1_i/m$, which is equal to the marginal empirical c.d.f. of the i-th feature value of node $u$.

Similarly, for every feature $i$ we have $k^2_i$. If there is another hypergraph node $(i, k', 2) \in \mathcal{V}(\mathcal{F}_\mathcal{E})$ such that $k' > k^2_i$, then let $n^2_i = k' - 1$. Otherwise, let $n^2_i = m$. Let $\boldsymbol{S}^H_{(i+d)(i+d)} = n^2_i/m$. Hence, we have filled in the diagonal entries of $\boldsymbol{S}^H$.

Second, we construct the off-diagonal entries of $\boldsymbol{S}^H$. Recall that the off-diagonal entries of STAGE-edge-graph weighted adjacency matrices denote the empirical conditional probabilities between two different features (Equation (2)), either within the same source node, the same target node, or between the source and target node. Specifically, for any two features $i, j \in [d]$, $i \neq j$, the entry is

$$\boldsymbol{S}^H_{ij} = \mathbb{P}_{A \sim \text{Unif}(V)}(\mathrm{x}^A_i \leq x^u_i \mid \mathrm{x}^A_j \leq x^u_j)$$
$$\boldsymbol{S}^H_{i(j+d)} = \mathbb{P}_{(A,B) \sim \text{Unif}(E)}(\mathrm{x}^A_i \leq x^u_i \mid \mathrm{x}^B_j \leq x^v_j)$$
$$\boldsymbol{S}^H_{(i+d)j} = \mathbb{P}_{(A,B) \sim \text{Unif}(E)}(\mathrm{x}^B_i \leq x^v_i \mid \mathrm{x}^A_j \leq x^u_j)$$
$$\boldsymbol{S}^H_{(i+d)(j+d)} = \mathbb{P}_{B \sim \text{Unif}(V)}(\mathrm{x}^B_i \leq x^v_i \mid \mathrm{x}^B_j \leq x^v_j)$$

where $(u, v)$ is the edge in the input graph corresponding to the hyperedge $H$.

We can compute these entries of $\boldsymbol{S}^H$ as follows. First, given any hyperedge $H' \in \mathcal{V}(\mathcal{F}_\mathcal{E})$, denote $K^d_{H'}(i)$ for any $i \in [d]$ and $r \in \{1, 2\}$ such that $(i, K^d_{H'}(i), r) \in H'$. Then, regarding our particular hyperedge $H$ of interest, for every pair of features $i, j \in [d]$ with $i \neq j$, we can obtain $n^1_i, n^1_j, n^2_i$, and $n^2_j$ as defined previously. Recall that $n^d_i$ is the number of feature values of i-th feature that are smaller than or equal to the current i-th feature value captured by $H$, for both the source node ($r = 1$) or the target node ($r = 2$).

For the entry $\boldsymbol{S}_{ij}^H$ and $\boldsymbol{S}_{(i+d)(j+d)}^H$, they capture inner-node feature dependencies, and we notice that the empirical conditional probabilities are defined w.r.t. random nodes sampled uniformly from the set of all nodes. Hence, we can compute these two entries as follows:

$$\boldsymbol{S}_{ij}^H = \min\{1, n_i^1/n_j^1\}$$
$$\boldsymbol{S}_{(i+d)(j+d)}^H = \min\{1, n_i^2/n_j^2\}.$$

To compute the entries for $\boldsymbol{S}_{i(j+d)}^H$ and $\boldsymbol{S}_{(i+d)j}^H$, we note that the random nodes A, B are uniformly sampled from the set of edges $E$. To do so, we first define the two subsets of hyperedges $\mathcal{H}_j^1$ and $\mathcal{H}_j^2$ as follows:

$$\mathcal{H}_j^1 := \{K_{H'}^1(j) \le n_j^1 \mid H' \in \mathcal{H}(\mathcal{F}_\mathcal{E})\}$$
$$\mathcal{H}_j^2 := \{K_{H'}^2(j) \le n_j^2 \mid H' \in \mathcal{H}(\mathcal{F}_\mathcal{E})\}.$$

In other words, $\mathcal{H}_j^1$ is the subset of hyperedges whose node, $(i, K_{H'}^1(j), 1)$, has an order $K_{H'}^1(j)$ that is smaller than or equal to the order of the counterpart node of the current hyperedge $H$. Vice versa for $\mathcal{H}_j^2$. Hence, we have

$$|\mathcal{H}_j^1|/|\mathcal{H}(\mathcal{F}_\mathcal{E})| = \mathbb{P}_{(A,B)\in\mathrm{Unif}(E)}(\mathrm{x}_j^A \le x_j^u)$$
$$|\mathcal{H}_j^2|/|\mathcal{H}(\mathcal{F}_\mathcal{E})| = \mathbb{P}_{(A,B)\in\mathrm{Unif}(E)}(\mathrm{x}_j^B \le x_j^v).$$

Then, we define the next two subsets $\mathcal{H}_{i|j}^{1|2}$ and $\mathcal{H}_{i|j}^{2|1}$ as follows:

$$\mathcal{H}_{i|j}^{1|2} := \{K_{H'}^1(i) \le n_i^1 \mid H' \in \mathcal{H}_j^2\}$$
$$\mathcal{H}_{i|j}^{2|1} := \{K_{H'}^2(i) \le n_i^2 \mid H' \in \mathcal{H}_j^1\}$$

These two subsets help us effectively computes the empirical conditional probabilities. Namely, now we have

$$|\mathcal{H}_{i|j}^{1|2}|/|\mathcal{H}_j^2| = \mathbb{P}_{(A,B)\in\mathrm{Unif}(E)}(\mathrm{x}_i^A \le x_i^u \mid \mathrm{x}_j^B \le x_j^v)$$
$$|\mathcal{H}_{i|j}^{2|1}|/|\mathcal{H}_j^1| = \mathbb{P}_{(A,B)\in\mathrm{Unif}(E)}(\mathrm{x}_i^B \le x_i^v \mid \mathrm{x}_j^A \le x_j^u)$$

Thus, we set the adjacency matrix entries for inter-node dependencies to

$$\boldsymbol{S}_{i(j+d)}^H = |\mathcal{H}_{i|j}^{1|2}|/|\mathcal{H}_j^2|$$
$$\boldsymbol{S}_{(i+d)j}^H = |\mathcal{H}_{i|j}^{2|1}|/|\mathcal{H}_j^2|$$

Now that we have constructed a mapping $\mathcal{I}$ mapping $\mathbb{S}_E$ to $\mathcal{F}_\mathcal{E}$, and another mapping $\mathcal{I}^{-1}$ mapping $\mathcal{F}_\mathcal{E}$ to $\mathbb{S}_E$, we now want to check that they are valid bijections. To show this, we show that $\mathcal{I}^{-1} \circ \mathcal{I} = $ Identity, and $\mathcal{I} \circ \mathcal{I}^{-1} = $ Identity.

**Show that $\mathcal{I}^{-1} \circ \mathcal{I} = $ Identity**

Let $\mathbb{S}_E$ be an arbitrary multiset of STAGE-edge-graphs. Let $\mathcal{F}' = \mathcal{I}(\mathbb{S}_E)$ and $\mathbb{S}'' = \mathcal{I}^{-1}(\mathcal{F}') = \mathcal{I}^{-1}(\mathcal{I}(\mathbb{S}_E))$. First, we observe that the mapping $\mathcal{I}$ transforms each element $\boldsymbol{G}(\boldsymbol{S}^{uv}) \in \mathbb{S}_E$ to one hyperedge $H' \in F'$. Similarly, the mapping $\mathcal{I}^{-1}$ transforms each hyperedge $H' \in F'$ to one STAGE-edge-graph $\boldsymbol{G}'' \in \mathbb{S}''$. Hence, as long as we show that, for any $\boldsymbol{G}(\boldsymbol{S}^{uv}) \in \mathbb{S}_E$, the composed transformation $\mathcal{I}^{-1} \circ \mathcal{I}$ produces a STAGE-edge-graph $\boldsymbol{G}''$ such that $\boldsymbol{G}(\boldsymbol{S}^{uv}) = \boldsymbol{G}''$, we can conclude $\mathcal{I}^{-1} \circ \mathcal{I} = $ Identity.

To observe this, we first note that $\boldsymbol{G}''$ has the same set of labeled nodes with $\boldsymbol{G}$, and that each node $(i, r), i \in [d], r \in \{1, 2\}$ has the same empirical marginal c.d.f. values. Similarly, between any two nodes $(i_1, r_1)$ and $(i_2, r_2)$, $\boldsymbol{G}$ and $\boldsymbol{G}''$ will have the same edge attribute for the edge $((i_1, r_1), (i_2, r_2))$, which corresponds to the empirical conditional probabilities between features $i_1$ and $i_2$ and between node placement in the original edge (source or target) $r_1$ and $r_2$. Thus, $\boldsymbol{G} = \boldsymbol{G}''$.

**Show that $\mathcal{I} \circ \mathcal{I}^{-1} = $ Identity**

Let $\mathcal{F}_\mathcal{E}$ be an arbitrary feature hypergraph. Let $\mathbb{S}' = \mathcal{I}^{-1}(\mathcal{F}_\mathcal{E})$ and $\mathcal{F}'' = \mathcal{I}(\mathbb{S}')$. Similarly, as long as we show that, for any hyperedge $H \in \mathcal{F}_\mathcal{E}$, the composed transformation $\mathcal{I} \circ \mathcal{I}^{-1}$ produces a hypergraph $H''$ such that $H = H''$, we can conclude that $\mathcal{I} \circ \mathcal{I}^{-1} = $ Identity.

To observe this, we note that every hyperedge $(i, k, r) \in H$, where $i \in [d], 1 \leq k \leq m_i', r \in \{1, 2\}$, will be recovered in $H''$. This is because each $(i, k, r) \in H$ corresponds to a unique labeled node $(i, r)$ in the STAGE-edge-graph $\mathbb{G}'$, which will be used to construct a node $(i, k'', r)$ in $H''$ under the mapping $\mathcal{I}$. In terms of the order $k$, the mapping $\mathcal{I}^{-1}$ will converts it into the marginal empirical c.d.f. value, which is treated as the attribute of node $(i, r)$ in the STAGE-edge-graph $\mathbb{G}'$. The mapping $\mathcal{I}$, on the other hand, will convert this marginal empirical c.d.f. value into the order $k''$ for the node $(i, k'', r)$ in $H''$, guaranteeing $k'' = k$. Thus, every node $(i, k, r)$ that is in $H$ is also in $H''$, and there will be no additional nodes created for $H''$. Hence, $H = H''$ for every hyperedge $H \in \mathcal{F}_\mathcal{E}$, and thus $\mathcal{I} \circ \mathcal{I}^{-1} =$ Identity.

In conclusion, we have shown two mappings, $\mathcal{I}$ and $\mathcal{I}^{-1}$, and have shown that they are the inverse transformation of each other. Hence, $\mathcal{I}$ is a bijective mapping between the multiset of STAGE-edge-graphs and feature hypergraph. □

Given the bijective mapping in Lemma B.4 between the multiset of STAGE-edge-graphs with unique feature identifiers and the feature hypergraph, and the fact that the feature hypergraph allows for a maximal invariant graph representation (Lemma B.3), it follows that the set of STAGE-edge-graphs can also yield a maximal invariant representation of the original input graph. This observation is formalized as below, which is our second theoretical contribution:

**Theorem 3.3.** *Given the attributes of the endpoint nodes $\mathcal{E}$ (Definition 3.1) of a graph $G = (V, E, \boldsymbol{X})$, there exists an optimal parameterization $\theta_g^*, \theta_s^*$ for a most expressive GNN encoder $M^g$ and a most-expressive multiset encoder $M^s$, respectively, such that $M_{\theta_s^*, \theta_g^*}(G) := M_{\theta_s^*}^s(\{\{M_{\theta_g^*}^g(\boldsymbol{G}(\boldsymbol{S}^{uv})) : (u, v) \in E\}\})$ such that any test $T(\mathcal{E})$ that measures the dependence of $\mathcal{E}$'s attributes of the endpoint nodes has an equivalent function $h$ within the space of Multilayer Perceptrons (MLPs) that depends solely on the graph representation $M_{\theta_s^*, \theta_g^*}(G)$, i.e., $\exists h \in MLPs$ s.t. $T(\mathcal{E}) = h(M_{\theta_s^*, \theta_g^*}(G))$.*

*Proof.* To show invariance, let $G_1 = (V, E, \boldsymbol{X}_2)$ and $G_2 = (V, E, \boldsymbol{X}_2)$ be two graphs such that $f(\boldsymbol{X}_1) = \boldsymbol{X}_2$ for some groupoid action $f$ in the COGF. Let $\mathcal{E}_1$ and $\mathcal{E}_2$ be the corresponding attributes of the endpoint nodes respectively, from which we have $f(\mathcal{E}_1) = \mathcal{E}_2$. Let $\mathbb{S}_{1E} = \{\{\boldsymbol{G}(\boldsymbol{S}_1^{uv}) \mid (u, v) \in E\}\}$ and $\mathbb{S}_{2E} = \{\{\boldsymbol{G}(\boldsymbol{S}_2^{uv}) \mid (u, v) \in E\}\}$ be the corresponding STAGE-edge-graphs respectively.

Since $f(\mathcal{E}_1) = \mathcal{E}_2$, and the attribute hypergraph is invariant to COGF (shown in the proof for Lemma B.3), we have $\mathcal{F}_{\mathcal{E}_1} = \mathcal{F}_{\mathcal{E}_2}$. And since there is a one-to-one mapping between the multiset of STAGE-edge-graphs and the feature hypergraph, we have $\mathbb{S}_{1E} = \mathbb{S}_{2E}$. Hence,

$$\left\{\left\{M_{\theta_g^*}(\boldsymbol{S}_1^{uv}) \mid (u, v) \in E\right\}\right\} = \left\{\left\{M_{\theta_g^*}(\boldsymbol{S})\right\}\right\}_{\boldsymbol{S} \in \mathbb{S}_{1E}}$$
$$= \left\{\left\{M_{\theta_g^*}(\boldsymbol{S})\right\}\right\}_{\boldsymbol{S} \in \mathbb{S}_{2E}} = \left\{\left\{M_{\theta_g^*}(\boldsymbol{S}_2^{uv}) \mid (u, v) \in E\right\}\right\}.$$

As a result,

$$M_{\theta_s^*, \theta_g^*}(G_1) = M_{\theta_s^*}^s(\left\{\left\{M_{\theta_g^*}^g(\boldsymbol{G}(\boldsymbol{S}_1^{uv})) \mid (u, v) \in E\right\}\right\})$$
$$= M_{\theta_s^*}^s(\left\{\left\{M_{\theta_g^*}^g(\boldsymbol{G}(\boldsymbol{S}_2^{uv})) \mid (u, v) \in E\right\}\right\}) = M_{\theta_s^*, \theta_g^*}(G_2).$$

To show maximality, Let $G_1$ and $G_2$ be two graphs such that $M_{\theta_s^*, \theta_g^*}(G_1) = M_{\theta_s^*, \theta_g^*}(G_2)$. Then, because $M_{\theta_s^*}^s$ is a most expressive multiset encoder, we have that

$$M_{\theta_s^*}^s(\left\{\left\{M_{\theta_g^*}^g(\boldsymbol{G}(\boldsymbol{S}_1^{uv})) \mid (u, v) \in E\right\}\right\}) = M_{\theta_s^*}^s(\left\{\left\{M_{\theta_g^*}^g(\boldsymbol{G}(\boldsymbol{S}_2^{uv})) \mid (u, v) \in E\right\}\right\}).$$

Again, since $M_{\theta_g^*}^g$ is a most expressive GNN, we have

$$\mathcal{S}_{1E} = \{\{\boldsymbol{G}(\boldsymbol{S}_1^{uv}) \mid (u, v) \in E\}\} = \{\{\boldsymbol{G}(\boldsymbol{S}_2^{uv}) \mid (u, v) \in E\}\} = \mathcal{S}_{2E}.$$

This implies that the feature hypergraphs $\mathcal{F}_{\mathcal{E}_1}$ and $\mathcal{F}_{\mathcal{E}_2}$ are the same, $\mathcal{F}_{\mathcal{E}_1} = \mathcal{F}_{\mathcal{E}_2}$ due to the bijective mapping between multisets of STAGE-edge-graphs and feature hypergraphs. And as has been shown in the proof of Lemma B.3, this implies there exists a groupoid action $f$ in COGF such that $f(\mathcal{E}_1) = \mathcal{E}_2$. Hence, we have shown that $M_{\theta_s^*, \theta_g^*}(G)$ is a maximal invariant representation w.r.t. COGF.

Thus, similar to the proof of Theorem 3.2, there exists a MLP $h$ such that for any test $T(\mathcal{E})$, we have

$$T(\mathcal{E}) = h(M_{\theta_s^*, \theta_g^*}(G)).$$

□

## B.4. COGG Invariances

**Definition B.5** (Component-wise order-preserving groupoid for graphs (COGG)). Denote $\mathbb{X}$ the space of node attributes with $d \geq 1$ dimensions, and $\mathbb{G}(\mathbb{X})$ the space of attributed graphs with attribute space $\mathbb{X}$ and $m \geq 2$ entities. A graph transformation $g : \mathbb{G}(\mathbb{X}_1) \to \mathbb{G}(\mathbb{X}_2)$ of two attribute spaces $\mathbb{X}_1$ and $\mathbb{X}_2$ is said to be a groupoid action of the *component-wise order-preserving groupoid for graphs* if it can be decomposed into a permutation of node identities $g_{\text{node}} : V \to V$ and a transformation of node attributes $g_{\text{attribute}} : \mathbb{X}_1 \to \mathbb{X}_2$ satisfying the following. Given $G_1 = (V, E_1, \boldsymbol{X}_1) \in \mathbb{G}(\mathbb{X}_1)$ and $G_2 = (V, E_2, \boldsymbol{X}_2) \in \mathbb{G}(\mathbb{X}_2)$ with $g(G_1) = G_2$,

- $\forall u, v \in V, (u, v) \in E_1 \iff (g_{\text{node}}(u), g_{\text{node}}(v)) \in E_2$ .

- $g_{\text{attribute}}$ is a COGF (Definition B.2) except for any $i \in [d]$, the $i$-th component $g_{\text{attribute},i}$ may map the $i$-th dimension of $\mathbb{X}_1$ to a different dimension of $\mathbb{X}_2$, while maintaining a one-to-one correspondence between all dimensions of $\mathbb{X}_1$ and $\mathbb{X}_2$.

## B.5. STAGE as a COGG Invariant Representation

**Theorem 3.4.** *STAGE is invariant to COGGs (Definition B.5).*

*Proof.* Given a graph $G = (V, E, \boldsymbol{X})$, a STAGE model $M$ applies two instances of equivariant GNNs, an intra-edge GNN and an inter-edge one, to process the input graph. Denote the intra-edge GNN $M_1$ and the inter-edge GNN $M_2$. The intra-edge GNN $M_1$ is applied onto $\mathbb{S}_E := \{\{\boldsymbol{G}(\boldsymbol{S}^{uv}) \mid (u, v) \in E\}\}$, the set of STAGE-edge-graphs, to produce edge-leve embeddings:

$$\boldsymbol{r}^{uv} = M_1(\boldsymbol{G}(\boldsymbol{S}^{uv})), \forall (u, v) \in E$$

and the inter-edge GNN $M_2$ takes the edge-level embeddings as the edge attributes onto the original graph, i.e., making a $G' = \left(V, E, \left\{\left\{r^{uv}_{(u,v)\in E}\right\}\right\}\right)$ to produce a final graph representation:

$$M(G) = M_2(G') = M_2((V, E, \{\{r^{uv}\}\}_{(u,v)\in E}))$$

Now, consider a train graph $G_{\text{tr}} = (V_{\text{tr}}, E_{\text{tr}}, \boldsymbol{X}_{\text{tr}})$ with $\mathcal{E}_{\text{tr}}$ and a test graph $G_{\text{te}} = (V_{\text{te}}, E_{\text{te}}, \boldsymbol{X}_{\text{te}})$ such that there exists a groupoid action $g$ in the COGG (Definition B.5) satisfying $g(G_{\text{tr}}) = G_{\text{te}}$. As per Definition B.5, $g$ is composed of a node identity permutation $g_{\text{node}}$ and a attribute transformation $g_{\text{attribute}}$.

We first note that the multiset $\{\{r^{uv}\}\}_{(u,v)\in E}$ is invariant to node identity permutation $g_{\text{node}}$ because a multiset is invariant to the permutation of its elements. Since the inter-edge GNN $M_2$ is an equivariant GNN, we have that

$$M(g_{\text{node}}(G_{\text{tr}})) = M((g_{\text{node}}(V_{\text{tr}}), g_{\text{node}}(E_{\text{tr}}), \{\{r^{uv}\}\}_{(u,v)\in g_{\text{node}}(E_{\text{tr}})})) = M(G_{\text{tr}}).$$

Hence, as long as we can show that the graph representation given by $M$ is also invariant under $g_{\text{attribute}}$, then we together we can show that $M$ is invariant to our groupoid action $g$, and that $M(G_{\text{tr}}) = M(G_{\text{te}})$.

To proceed, we first note that the groupoid action $g_{\text{attribute}}$, when applied to an attributed graph $G$, can be expressed as $g_{\text{attribute}}(G) = (V, E, g_{\text{attribute}}(\boldsymbol{X}))$, because the attribute transformation only acts on the node attributes but leaves the graph structure unchanged. Hence, when applying the inner-edge GNN $M_1$ to the multiset of STAGE-edge-graphs of a transformed input graph $g_{\text{attribute}}(G)$, we write $M_1(g_{\text{attribute}}(\boldsymbol{G}(\boldsymbol{S}^{uv})))$, for all $(u, v) \in E$.

Now, all we need to show is that the intra-edge GNN $M_1$ produces a multiset of STAGE-edge-graph representations that is invariant under the attribute transformation $g_{\text{attribute}}$, i.e., $\{\{M_1(\boldsymbol{G}(\boldsymbol{S}^{uv}))\}\}_{(u,v)\in E} = \{\{M_1(g_{\text{attribute}}(\boldsymbol{G}(\boldsymbol{S}^{uv})))\}\}_{(u,v)\in E}$. Since $g_{\text{attribute}}$ is COGF (Definition B.2) except it may map different training attribute dimensions of $\boldsymbol{X}_{\text{tr}}$ to different attribute dimensions of $\boldsymbol{X}_{\text{te}}$, we can therefore further decompose it into two different components: $h$ and $f$ with $g = h \circ f$, where $h$ is a mapping that permutes attribute dimensions, and $f$ is a COGF.

In Theorem 3.3, we have shown that a most expressive GNN applied to a STAGE-edge-graph $\boldsymbol{G}(\boldsymbol{S}^{uv})$ equipped with attribute ids (which are the nodes ids in the STAGE-edge-graph because nodes correspond to attribute dimensions) produces maximal invariant representation under COGF. Hence, this implies that the intra-edge GNN $M_1$, when applied to each STAGE-edge-graph, without unique node ids, will produce an invariant representation to the COGF $f$. Namely, for all $(u, v) \in E$,

$$M_1(f(\boldsymbol{G}(\boldsymbol{S}^{uv}))) = M_1(\boldsymbol{G}(f(\boldsymbol{S}^{uv}))) = M_1(\boldsymbol{G}(\boldsymbol{S}^{uv})).$$

Note that $f(\boldsymbol{G}(\boldsymbol{S}^{uv})) = \boldsymbol{G}(f(\boldsymbol{S}^{uv}))$ because $f$ acts on the node and edge attributes in $\boldsymbol{G}(\boldsymbol{S}^{uv})$ (which are derived from the attribute values), but preserve the graph structure.

On the other hand, once the node ids in STAGE-edge-graph $\boldsymbol{G}(\boldsymbol{S}^{uv})$ is dropped, because $M_1$ is an equivariant GNN, we also have that the $M_1$'s output representations are invariant to permutations of the attribute dimensions, which corresponds to the permutations of node ids in the STAGE-edge-graph. Namely, for all $(u, v) \in E$,

$$M_1(h(\boldsymbol{G}(\boldsymbol{S}^{uv}))) = M_1(\boldsymbol{G}(\boldsymbol{S}^{uv})).$$

Hence, together we have that for any $(u, v) \in E$,

$$M_1(g_{\text{attribute}}(\boldsymbol{G}(\boldsymbol{S}^{uv}))) = M_1(h \circ f(\boldsymbol{G}(\boldsymbol{S}^{uv}))) = M_1(h(\boldsymbol{G}(\boldsymbol{S}^{uv}))) = M_1(\boldsymbol{G}(\boldsymbol{S}^{uv})),$$

Thus completing the proof. $\qquad\square$

## C. Datasets

Here we describe how we construct the E-Commerce Category Dataset, the H&M Dataset, and the Social Network Datasets (Friendster and Pokec).

### C.1. E-Commerce Category Dataset

To test the model's generalization to new input attribute spaces, we consider a dataset of E-Commerce users and products (Kechinov, 2020). There are 29,228,809 different product categories, such as smartphones, shoes, and computers. We select a subset of the most popular product categories and form an input graph from the products under each category and their respective connected users. At test time, we hold out an entirely different graph containing unseen products, *from new unseen categories* and associated users, and test the zero-shot (i.e., frozen model) performance on the test data. In this dataset, we focus on the single task of predicting links between users and products, with links indicating a user purchasing/viewing/carting/uncarting a product.

However, all categories originally share the same attributes. To ensure that the graph domains we build have different attribute types, we use GPT-4 to retrieve information specific to each category. Specifically, the information retrieval process involves prompting GPT-4 with the following content:

```
"According to the following information regarding an E-Commerce purchase, give
    information about the product in the following asked format."
"First, the product is purchased at time: " + row["event_time"] + "."
"Second, the category of the product is " + row["category_code"] + "."
"Third, the brand of the product is " + row["brand"] + "."
"Last, the price of the product is " + str(row["price"]) + "."
"Please provide information about the product in the following json format."
"{json_prototype}"
```

The JSON prototype is different for different categories, and contains attributes that are specific for the category being prompted. That is, the JSON prototype for smartphones contains, for instance, attributes like *display type*, which is not a attribute for shoes, containing instead attributes such as *ankle height*. In the following, we report the JSON prototype for all categories.

```
bed

{
    "type": <select from ['Twin', 'Twin XL', 'Full', 'Queen', 'King', 'California
        King']>,
    "material": <select from ['Wood', 'Metal', 'Upholstered', 'Bamboo', 'Particle
        Board', 'Composite']>,
    "bed_frame_included": <select from ['True', 'False']>,
```

```
    "headboard_included": <select from ['True', 'False']>,
    "footboard_included": <select from ['True', 'False']>,
    "mattress_included": <select from ['True', 'False']>,
    "box_spring_required": <select from ['True', 'False']>,
    "weight_capacity_lbs": <give int in lbs>,
    "bed_size_length_inches": <give float in inches>,
    "bed_size_width_inches": <give float in inches>,
    "bed_size_height_inches": <give float in inches>
}
```

**desktop**

```
{
    "processor_type": <select from ['Intel Core i3', 'Intel Core i5', 'Intel Core i7
        ', 'Intel Core i9', 'AMD Ryzen 3', 'AMD Ryzen 5', 'AMD Ryzen 7', 'AMD Ryzen
        9', 'Apple M1', 'ARM other']>,
    "ram_gb": <give int>,
    "storage_type_hdd_size_gb": <give int>,
    "storage_type_ssd_size_gb": <give int>,
    "storage_type_hybrid_size_gb": <give int>,
    "graphics_card": <select from ['NVIDIA GeForce GTX 1660', 'NVIDIA GeForce RTX
        2060', 'NVIDIA GeForce RTX 2070', 'NVIDIA GeForce RTX 2080', 'AMD Radeon RX
        570', 'AMD Radeon RX 580', 'AMD Radeon RX 590', 'AMD Radeon RX 5700', 'AMD
        Radeon RX 5700 XT']>,
    "operating_system": <select from ['Windows 10', 'macOS', 'Linux Ubuntu', 'Linux
        Fedora', 'Linux Mint', 'Debian', 'FreeBSD']>,
    "power_supply_watts": <give int>,
    "cooling_system": <select from ['Air cooling', 'Liquid cooling', 'Passive
        cooling']>,
    "has_bluetooth": <select from ['True', 'False']>
}
```

**refrigerators**

```
{
    "energy_rating": <select from ['A+++', 'A++', 'A+', 'A', 'B', 'C']>,
    "capacity_liters": <give int>,
    "refrigerator_type": <select from ['Top Freezer', 'Bottom Freezer', 'Side-by-
        Side', 'French Door', 'Mini Fridge', 'Commercial']>,
    "defrost_type": <select from ['Manual', 'Frost Free', 'Automatic Defrost']>,
    "has_ice_maker": <select from ['True', 'False']>,
    "has_water_dispenser": <select from ['True', 'False']>,
    "has_smart_technology": <select from ['True', 'False']>,
    "is_energy_efficient": <select from ['True', 'False']>,
    "height_cm": <give float>,
    "width_cm": <give float>,
    "depth_cm": <give float>
}
```

**smartphone**

```
{
    "display_type": <select from ['OLED', 'LCD']>,
    "display_size": <give float in inches>,
    "display_resolution": <give int in pixels>,
    "processor_type": <give string>,
```

```
    "ram": <give int in GB>,
    "storage_options": <give int in GB>,
    "rear_camera_primary_resolution": <give int in MP>,
    "front_camera_resolution": <give int in MP>,
    "operating_system": <select from ['Android', 'iOS', 'HarmonyOS', 'KaiOS', 'Tizen
        ', 'Ubuntu Touch', 'PureOS', 'Sailfish OS', 'Plasma Mobile']>,
    "Battery_capacity": <give int in mAh>,
    "Has_gps":  <select from ['True', 'False']>,
    "has_nfc":  <select from ['True', 'False']>
    }
```

**shoes**

```
{
    "type": <select from ['Running', 'Casual', 'Formal', 'Sports', 'Boots', 'Sandals
        ', 'Slippers', 'Hiking', 'Dress', 'Work', 'Safety']>,
    "material": <select from ['Leather', 'Synthetic', 'Textile', 'Rubber', 'Canvas',
        'Mesh', 'Suede', 'Patent Leather', 'Nubuck', 'Faux Leather']>,
    "color": <give string>,
    "size": <give float in UK sizes>,
    "gender": <select from ['Men', 'Women', 'Unisex', 'Children', 'Infants']>,
    "closure_type": <select from ['Laces', 'Velcro', 'Slip-on', 'Buckle', 'Zip', '
        Hook and Loop', 'None']>,
    "sole_material": <select from ['Rubber', 'Synthetic', 'PVC', 'EVA', 'Leather', '
        TPU (Thermoplastic Polyurethane)', 'TPR (Thermoplastic Rubber)']>,
    "water_resistant": <select from ['True', 'False']>,
    "ankle_height": <select from ['Low-top', 'Mid-top', 'High-top', 'Over the ankle
        ']>,
    "breathability": <select from ['High', 'Medium', 'Low']>,
    "weight": <give float in grams>,
    "origin_country": <give string>,
    "seasonality": <select from ['All-season', 'Summer', 'Winter', 'Rainy', 'Spring
        ', 'Autumn']>,
    "eco_friendly": <select from ['True', 'False']>
}
```

After extracting attributes of different numbers for all categories, we also append the original two shared attributes of all products (*price, brand*) that are considered to have a different distribution across categories, forming the following dataset statistics. Since the customer nodes lack attributes, we build edges between attributed nodes of the same type (e.g., products) based on common connections, forming *STAGE-edge-graph* for these new edges. These edges are provided to all baselines.

Table 3: Statistics of E-Commerce Categories

| Category | Number of Nodes | Number of Edges | Average Degree | Num attributes |
|----------|-----------------|-----------------|----------------|----------------|
| **bed** | 4044 | 25788 | 6.38 | 13 |
| **desktop** | 3011 | 37450 | 12.44 | 12 |
| **refrigerators** | 2985 | 33520 | 11.23 | 13 |
| **smartphone** | 3391 | 31970 | 9.43 | 14 |
| **shoes** | 4032 | 54890 | 13.62 | 16 |

### C.2. H&M Dataset

H&M has 106K products, sharing the same 25 attributes, and 1.37M customers, sharing the same 7 attributes. We sampled the interaction between the most popular 830 products and 830 customers based on their node degrees. We discarded 14 product attributes since 12 of them are repetitive (e.g. *perceived_colour_value_id* is just a one-to-one mapping of *perceived_colour_value_name*), 1 of them is the *detail_desc* an English sentence that connects the other attributes, and 1

Table 4: Comparison of statistics between the Pokec and Friendster social network datasets after filtering and sampling nodes.

| Statistics | Pokec | Friendster |
|---|---|---|
| Number of nodes | 283 | 1392 |
| Number of edges | 2084 | 3322 |
| Number of node attributes | 4 | 5 |
| Attributes | public, completion_percentage,region, age | age, interest, occupation,music, tv |
| Average degree | 7.363957597173145 | 2.3864942528735633 |
| Minimum degree | 1 | 1 |
| Maximum degree | 29 | 12 |
| Lowest degrees | [1, 2, 3, 4, 5] | [1, 2, 3, 4, 5] |
| # Nodes with lowest degrees | [35, 31, 25, 21, 14] | [516, 404, 213, 113, 62] |
| Label 0 Ratio | 50.88% | 46.84% |
| Label 1 Ratio | 49.12% | 53.16% |

of them is the *article_id* serving as the identifier of each product. We also discarded 4 user attributes: *customer_id* as the identifier, *FN* and *Active* due to too many missing values (65% and 66% respectively), and *postal_code* that is overdispersed.

After picking the largest connected component of the graph formed by the 830 products and 830 users, we construct this dataset to have 77080 edges, 1580 nodes with an average degree 48.78, and 11 attributes for each product node and 3 attributes for each user node. The product attributes are: *product_type_name*, *product_group_name*, *graphical_appearance_name*, *colour_group_name*, *perceived_colour_value_name*, *perceived_colour_master_name*, *department_name*, *index_name*, *index_group_name*, *section_name*, *garment_group_name*. The user attributes are: *club_member_status*, *fashion_news_frequency*, *age*.

### C.3. Social Network Datasets (Friendster and Pokec)

The original Pokec social network dataset contains 1632803 nodes and 30622564 edges and each node has 58 attributes. However, 54 of them are difficult to encode either because they are random texts input by the user or because there is no straightforward way to turn the attributes into totally ordered ones. We first filtered out the nodes that contain invalid attributes and then sample the most popular 150 female and male nodes each before picking the largest connected components of the graph formed by the popular nodes.

The original Friendster social network dataset contains 43880 nodes and 145407 edges and each node has 644 attributes. However, most of the attributes are binary, which is inefficient for STAGE to encode (i.e. will need 644*2 nodes in each STAGE-edge-graphs). We find out the attributes are in the format of a meta attribute (e.g. occupation) followed by a more detailed attribute (e.g. writer). Therefore, we turned the binary attributes that share the same meta attribute into a multicategorical attribute. We then filtered out the nodes that have only one active binary attribute under each meta attribute (otherwise the multi-category does not make sense) and pick the largest connected components of the graph formed by the these nodes.
In the end, the statistics of Pokec and Friendster datasets are available in Table 4.

## D. Experiment Details

For Figure 3, Table 1, and Figure 4 We use the default NBFNet-PyG configuration for the inductive WN18RR dataset (Zhu et al., 2021c), except for a few specific parameters. The input dimension for the node attribute is set to 256, and the model includes six hidden layers with dimensions [256, 256, 256, 256, 256, 256], making a total of seven layers. For STAGE, we use 1 layer of GINEConv (Hu et al., 2020) for the GNN on *STAGE-edge-graph*, which produces an edge representation of dimension 256. We also append an extra p_value to each edge in the *STAGE-edge-graph* for expressivity. All model are trained with a batch size of 32 over 30 epochs.

For Figure 3, Figure 4, and the E-Commerce columns of Table 1 we average over seeds 0, 1, 2. For the H&M columns of Table 1, we average over seeds 1024, 1025, 1026.

For Table 2, we average over seeds 32, 33, and 34 using the following configuration. The input attribute dimension is set to 64, with 128 as the dimension of hidden channels. The model uses 2 layers of GINEConv (Hu et al., 2020). The learning

Table 5: Zero-shot test Mean-Square Loss (lower is better) of STAGE and baselines on the Pokec dataset with regression tasks on predicting the user's age. Models were trained on the same sample of the Friendster dataset in Section 4. All models show the same bad performance on doing this very challenging task because the root mean squared error (RMSE) of constantly predicting the mean of all age values is 10.7. We use the same configurations as Table 2. N/A means the model does not support node regression tasks.

| Model | RMSE ($\downarrow$) |
|---|---|
| GINE-structural | $10.99 \pm 0.000$ |
| GINE-gaussian | $10.99 \pm 0.000$ |
| GINE-normalized | $10.99 \pm 0.000$ |
| GINE-llm | $10.99 \pm 0.000$ |
| GINE-age | $10.99 \pm 0.000$ |
| GraphyAny | N/A |
| GCOPE | N/A |
| **GINE-STAGE (Ours)** | **$10.99 \pm 0.000$** |

rate for the optimizer was set to 0.0001, with a dropout rate of 0.5 to mitigate overfitting. Training was carried out for 400 epochs. Additionally, STAGE is deployed with 2 layers of GNN on *STAGE-edge-graph* with GINEConv and an edge representation of dimension 32. For GraphAny, we adopt the default configuration as preliminary experiments indicated that modifying hyperparameters yielded no significant performance improvements.

## E. Age Regression Experiment Results

Table 5 shows that the zero-shot regression on age across different social networks is a challenging task, particularly when the age distributions of the datasets are drastically different. GraphAny and GCOPE are not included because they are designed for and only supports node classification tasks. Figure 5 shows that the age distribution in the Pokec dataset is skewed towards younger users, with notable frequencies for ages such as 0 (invalid data), 18, and 20, while ages above 42 are scarcely represented. In contrast, the Friendster dataset contains a much broader range of ages, including significant numbers of users aged in their mid-twenties, such as 25, with smaller frequencies for users up to age 91. This disparity in distribution—where Pokec's frequencies are centered around younger users and Friendster's are more spread across the adult age spectrum—poses a substantial difficulty for models attempting to generalize across the two networks.

## F. Ablation Study

In this section, we provide ablation studies to further investigate the effectiveness and versatility of STAGE. Experiments in Appendix F.1 complement the main results in the paper by exploring whether STAGE is effective on alternative GNN backbones and configurations. Experiments in Appendix F.2 then study if STAGE can outperform a model trained on the common attributes shared between train and test domain, validating whether STAGE truly leverages dependencies among unseen attributes at test time to make predictions.

### F.1. Evaluating STAGE with GCN as the backbone GNN

In the main experiments, we employed GINE + NBFNet for link prediction and GINE + GINE for node classification as the backbone GNN configurations. A natural question arises: Can STAGE be effective when using other backbone GNN architectures? To address this, we propose GCN-STAGE (GINE + GCN (Kipf & Welling, 2016)), where we replace the second GINE with a modified GCN to perform message passing on the original graph. We choose GCN as it is a well-known baseline for node classification tasks. We modified GCN to process edge attributes by applying an MLP layer to edge attributes before incorporating them into the edge messages. The first GINE model operating on STAGE-edge-graphs remained unchanged.

Table 6 presents the results, which demonstrate that GCN-STAGE outperforms all baseline methods in terms of average accuracy. Comparing to the other GCN-backbone models, GCN-STAGE outperforms with a 7.33% relative improvement, and achieves an order-of-magnitude smaller standard deviation, showcasing the stability and consistency of predictions across random seeds. Furthermore, same as GINE-STAGE, GCN-STAGE also outperforms GraphAny (Zhao et al., 2024b), demonstrating that STAGE is effective on both GCN and GINE. We note that, however, the gain observed with GCN-STAGE

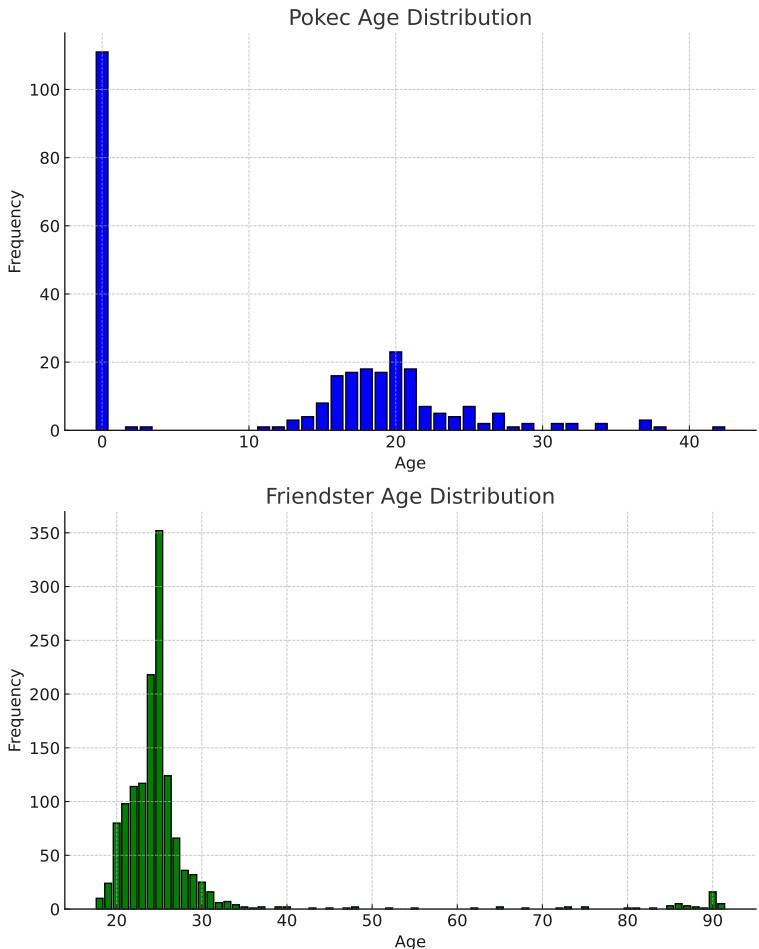

Figure 5: Comparison of Age Distributions in Pokec and Friendster Datasets. The top histogram shows the age distribution for the Pokec dataset, where a significant number of users have an age of 0, followed by a noticeable peak around the age of 20. The bottom histogram illustrates the age distribution for the Friendster dataset, with a strong concentration of users around the age of 25, and a smaller presence of older individuals.

is slightly lower than that of GINE-STAGE as shown in Table 2. This is not surprising, as GCN has been shown to have lesser expressivity than GINE (Xu et al., 2018).

These results demonstrate the effectiveness of STAGE regardless of the backbone GNN architecture (GINE or GCN), reinforcing the versatility and general applicability of STAGE across tasks and architectures, further solidifying its strength as a robust framework.

### F.2. Comparison with models trained on common attributes

In the second ablation study, we aim to investigate whether STAGE is truly leveraging dependencies among multiple unseen node attributes to make zero-shot predictions on the test domain, rather than simply relying on the common attributes shared between train and test. In particular, the attribute "price" and "brand" are shared between the E-commerce datasets (Appendix C.1), and the attribute "age" is shared between Friendster and Pokec (Appendix C.3). Hence, we compare STAGE to a model with the same backbone GNN trained to utilize the shared attribute to make predictions. We name these models NBFNet-price on E-commerce datasets for link prediction, and GINE-age on Friendster and Pokec for node classification. We do not experiment with training on the "brand" attribute because its values are distinct (or the distribution have different supports) in different product categories.

Table 6: Zero-shot test accuracy (higher is better) of STAGE and baselines on the Pokec dataset. Models were trained on a sample of the Friendster dataset. **GCN-STAGE demonstrates the best zero-shot test accuracy, surpassing all other methods in both average accuracy and stability.**

| Model | Accuracy ($\uparrow$) |
|---|---|
| GCN-structural | $0.547 \pm 0.0658$ |
| GCN-gaussian | $0.567 \pm 0.0382$ |
| GCN-normalized | $0.570 \pm 0.0315$ |
| GCN-llm | $0.526 \pm 0.0300$ |
| GraphAny | $0.591 \pm 0.0083$ |
| GCOPE | $0.535 \pm 0.0153$ |
| **GCN-STAGE (Ours)** | $\mathbf{0.593 \pm 0.0046}$ |

Table 7: Zero-shot Hits@1 and MRR of NBFNet-STAGE and NBFNet-price on the E-Commerce dataset. Models are trained on all combinations of four graph domains and tested on the remaining domain. **NBFNet-STAGE significantly outperforms NBFNet-price, demonstrating that STAGE effectively utilizes more information than common attribute (price) shared between attribute domains.**

| Model | Hits@1 ($\uparrow$) | MRR ($\uparrow$) |
|---|---|---|
| NBFNet-price | $0.2713 \pm 0.0280$ | $0.3263 \pm 0.0301$ |
| **NBFNet-STAGE (Ours)** | $\mathbf{0.4606 \pm 0.0123}$ | $\mathbf{0.4971 \pm 0.0073}$ |

Tables 7 and 8 shows the results of this ablation study. NBFNet-STAGE outperforms NBFNet-price with a relative improvement of 69.8% and GINE-STAGE outperforms GINE-age with a relative improvement of 12.0%. These results corroborates our statement that STAGE is capable of leveraging complex dependencies among multiple attributes to make predictions, even when said attributes are unseen during training, as STAGE significantly outperforms the models relying only on shared attributes.

## G. Interpreting STAGE

In Section 4, we demonstrated STAGE has a strong performance when zero-shotting to unknown attribute domains. A natural question arises: how does STAGE recognize unseen attributes during zero-shot testing, and which attributes are most relevant for making predictions? To address this, we conduct a qualitative analysis of STAGE's behavior using saliency maps (Erhan et al., 2009; Simonyan et al., 2013). This method computes the gradients of the model's outputs with respect to the input data, quantifying how much each input influences the model's prediction.

We train NBFNet-STAGE on all 5 categories of the E-Commerce dataset. We then perform zero-shot inference on the H&M dataset, obtaining triplet scores for missing edges. Let $G = (V, E, \mathbf{X})$ be the input graph of the H&M dataset with $d$ attributes, $\mathcal{T}$ be the set of ground-truth triplets in H&M, and $M(t), t \in \mathcal{T}$ be the model's output triplet score. Recall that STAGE transforms raw attribute values into edge attributes $\mathbf{S}^{uv}_{f_1 f_2}, \mathbf{S}^{uv}_{(f_1+d)f_2}, \mathbf{S}^{uv}_{f_1(f_2+d)}, \mathbf{S}^{uv}_{(f_2+d)(f_1+d)}$ for each pair of attributes $f_1, f_2$ and every edge $(u,v) \in E$. We compute the saliency of attribute pairs, $E_{G,\mathcal{T}}(f_1, f_2)$, as follows:

$$E_{G,\mathcal{T}}(f_1, f_2) \coloneqq \sum_{t \in \mathcal{T}} \sum_{(u,v) \in E} \left| \frac{\partial M(t)}{\partial \mathbf{S}^{uv}_{f_1 f_2}} + \frac{\partial M(t)}{\partial \mathbf{S}^{uv}_{f_1(f_2+d)}} + \frac{\partial M(t)}{\partial \mathbf{S}^{uv}_{(f_2+d)(f_1+d)}} + \frac{\partial M(t)}{\partial \mathbf{S}^{(f_1+d)(f_2+d)}_{ij}} \right.$$
$$\left. + \frac{\partial M(t)}{\partial \mathbf{S}^{uv}_{f_2 f_1}} + \frac{\partial M(t)}{\partial \mathbf{S}^{uv}_{f_2(f_1+d)}} + \frac{\partial M(t)}{\partial \mathbf{S}^{uv}_{f_2(f_1+d)}} + \frac{\partial M(t)}{\partial \mathbf{S}^{uv}_{(f_2+d)(f_1+d)}} \right|.$$

Thus, $E_{G,\mathcal{T}}(f_1, f_2)$ indicates how *the pair of attributes $f_1, f_2$ jointly influence* the model's output predictions.

One of the most striking aspects of STAGE is its ability to recognize relevant attribute dependencies zero-stho at inference time (i.e., without requiring fine-tuning). To investigate this phenomenon, we examine the saliency values of every pair of product attributes $f_1, f_2$ in the H&M dataset during zero-shot inference.

Figure 6 presents a heatmap of these normalized saliency values $E_{G,\mathcal{T}}(f_1, f_2)$. The color bar indicates that lighter shades

Table 8: Zero-shot test accuracy of GINE-STAGE and GINE-age on the social network datasets. Models are trained on Friendster and zero-shot tested on Pokec. **GINE-STAGE outperforms GINE-age, demonstrating that STAGE effectively utilizes more information than common attribute (age) shared between attribute domains.**

| Model | Accuracy (↑) |
|---|---|
| GINE-price | $0.582 \pm 0.0657$ |
| **GINE-STAGE (Ours)** | **$0.652 \pm 0.0042$** |

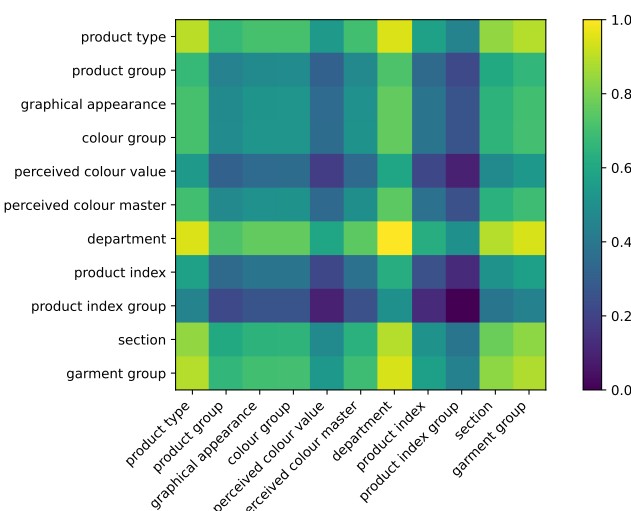

Figure 6: Saliency heatmap of the product attribute pairs in the H&M dataset. A lighter color indicates a larger gradient of the model's outputs w.r.t. the pair of attributes, hence the attribute pair is more relevant to the model's predictions. **attributes that describes the product's type and categories – "product type", "department", "section", "garment group" – generally are more relevant, whereas product indexes – "product index", "product index group" – are least relevant to the model's predictions.**

correspond to larger saliency values, signifying a greater impact on the model's output. Surprisingly, we find that certain attribute pairs exhibit high saliency values, such as "product type", "department", "section", and "garment group" because these attributes form a natural hierarchical taxonomy that effectively narrows down and defines specific products. A product's identity is progressively constrained through this hierarchy, from department (e.g., Ladies), to section (e.g., Clothes), to garment group (e.g., Dresses), to specific product type (e.g., Cocktail Dress). Each level in this taxonomy provides increasingly specific product categorization, making these attribute pairs particularly informative for product identification and classification. This is notable because these attributes are not explicitly labeled or weighted in our dataset; instead, STAGE has learned to recognize their importance while pretained on the different attribute space of E-commerce.

In contrast, attribute pairs like "product index" and "product index group" demonstrate low saliency values. This makes sense, as these attributes are arbitrary numerical identifiers that carry no semantic meaning about the product's characteristics, intended use, or target demographic. Unlike meaningful attributes that describe product properties, these index values are simply database artifacts used for internal record-keeping. However, the fact that STAGE can distinguish between relevant and irrelevant attributes without explicit guidance is a testament to its ability to capture subtle patterns in the data.

These results are particularly remarkable because they emerge from zero-shot inference, where all attributes are unseen and no fine-tuning has been performed. This suggests that STAGE is capable of generalizing to new domains and tasks, even when faced with unfamiliar attribute sets. Our findings reinforce our main experimental results in Section 4, providing further evidence of STAGE's ability to capture pertinent attribute relationships for effective task performance zero-shot.

Table 9: Average per-epoch training and inference time on E-Commerce dataset and zero-shot Hits@1 performance of NBFNet-STAGE and baselines on H&M dataset. Time is measured on an 80GB A100 GPU and averaged across 3 training epochs. **NBFNet-STAGE is 7.83% slower in training than the fastest baseline NBFNet-llm and 17.02% slower in inference than the fastest baseline NBFNet-raw, while being respectively 103% and 933% better**.

| Model | Wall Time per Training Epoch (s) | Inference Time on Test (s) | Zero-shot Hits@1 on H&M |
|---|---|---|---|
| NBFNet-raw | 318.65 | 66.16 | $0.0005 \pm 0.0004$ |
| NBFNet-gaussian | 322.13 | 69.04 | $0.0925 \pm 0.0708$ |
| NBFNet-structural | 322.31 | 69.78 | $0.2231 \pm 0.0060$ |
| NBFNet-llm | 316.55 | 67.57 | $0.2302 \pm 0.0015$ |
| NBFNet-normalized | 316.87 | 68.36 | $0.2286 \pm 0.0010$ |
| **NBFNet-STAGE (Ours)** | 341.36 | 77.42 | $\mathbf{0.4666 \pm 0.0020}$ |

## H. Complexity Analysis and Runtime Comparison

### H.1. STAGE Time Complexity

Here we analyze the time complexity of STAGE. In particular, we analyze NBFNet-STAGE (used for link prediction) and GINE-STAGE (used for node classification).

Let $d$ be the number of attributes, $h$ the dimension of internal node and edge embeddings, $|E|$ the number of edges, and $|V|$ the number of nodes in the input graph. For all tasks, STAGE consists of three steps:

1. **Fully Connected STAGE-edge-graph Construction:** This step requires $O(|E|d^2)$ operations because each fully connected STAGE-edge-graph has $2d$ nodes, and each edge in the original graph induces a fully connected STAGE-edge-graph.

2. **Inference on STAGE-edge-graphs:** We use 2 shared layers of GINE for all STAGE-edge-graphs. A single layer on one fully connected STAGE-edge-graph has complexity $O(dh + d^2h) = O(d^2h)$, since we have $2d$ $h$-dimensional nodes and $(2d)^2$ $h$-dimensional edges in each STAGE-edge-graph. Obtaining edge embeddings across all STAGE-edge-graphs takes $O(|E|d^2h)$.

3. **Inference on the original graph:** For link prediction tasks, we use NBFNet to perform message passing on the original graph, which requires $O(|E|h + |V|h^2)$ for one forward pass (where one forward pass gives representations conditioned on a single source node and relation, therefore, predicting links (s, q, ?) for a given source node s and relation q) (Zhu et al., 2021c). For node classification tasks, we use GINE again, which requires $O(|E|h)$ time.

Hence, in total, running one forward pass has a complexity of $O(|E|d^2h + |E|h + |V|h^2)$ for NBFNet-STAGE, and $O(|E|d^2h + |E|h)$ for GINE-STAGE.

### H.2. Training and inference time comparison

The analysis above shows the theoretical complexity of STAGE. Now we study the computational overhead when deployed in practice. To this end, we measured the average wall time per training epoch of NBFNet-STAGE and GINE-STAGE on respectively the E-Commerce Stores dataset and the Friendster dataset (see Appendix C) as well as their average inference time on the H&M dataset and the Pokec dataset using an 80GB A100 GPU.

Tables 9 and 10 displays the runtime comparison results. We observe that in train NBFNet-STAGE is 7.83% slower than the fastest baseline (NBFNet-llm) and GINE-STAGE is 49.7% slower than the fastest baseline (GINE-gaussian), while being respectively 103% and 933% better in zero-shot Hit@1 on H&M. During inference, NBFNet-STAGE is 17.02% slower than the fastest baseline (NBFNet-raw) and GINE-STAGE is 61.8% slower than the fastest baseline (GINE-structural), while being respectively 11% and 16% better in zero-shot accuracy on Pokec. The additional time is due to computing STAGE-edge-graph embeddings during each forward pass, while building the STAGE-edge-graphs is a one-time pre-processing step. Moreover, Table 10 shows that GINE-STAGE achieves $1.74\times$ speedup in training and $4.01\times$ speedup in inference than the best baseline GraphAny, which is specifically designed for the same tasks.

Table 10: Average per-epoch training and inference time on Friendster and zero-shot accuracy of GINE-STAGE and baselines on Pokec. Time is measured on an 80GB A100 GPU and averaged across 20 training epochs. GINE-STAGE is 49.7% slower in training than the fastest baseline GINE-gaussian and 61.8% slower in inference than the fastest baseline GINE-structural, while being respectively 11% and 16% better. **Moreover, GINE-STAGE achieves 1.74× speedup in training and 4.01× speedup in inference than the best baseline GraphAny**.

| Model | Wall Time per Training Epoch (s) | Inference Time on Test (s) | Zero-shot Accuracy on Pokec |
|---|---|---|---|
| GINE-raw | 0.0313 | 0.0067 | $0.558 \pm 0.0829$ |
| GINE-gaussian | 0.0296 | 0.0061 | $0.588 \pm 0.0250$ |
| GINE-structural | 0.0292 | 0.0055 | $0.564 \pm 0.0466$ |
| GINE-llm | 0.0322 | 0.0064 | $0.550 \pm 0.0368$ |
| GINE-normalized | 0.0316 | 0.0072 | $0.541 \pm 0.0148$ |
| GraphAny | 0.0762 | 0.0357 | $0.591 \pm 0.0083$ |
| GCOPE | 1.0524 | 0.3619 | $0.535 \pm 0.0153$ |
| **GINE-STAGE (Ours)** | 0.0437 | 0.0089 | $\mathbf{0.652 \pm 0.0042}$ |

# I. Additional Discussion of Related Work

**Foundation Models for Graph Data.** Foundation models for graph data aim to create versatile graph models capable of generalizing across different graphs and tasks. Despite growing interest, achieving a truly universal graph foundation model remains challenging, especially due to the complexities in designing a suitable graph vocabulary that ensures transferability across datasets and tasks (Mao et al., 2024). Initial efforts in this direction convert attributed graphs into texts and apply an LLM, but this methodology, while promising, risks information loss and may limit transferability (Collins et al., 2024; Gruver et al., 2024; Schwartz et al., 2024). For instance, OFA (Liu et al., 2024) uses frozen LLMs to generate attributes, and then trains a GNN to perform multiple tasks, while Chen et al. (2024b); Fatemi et al. (2024); Perozzi et al. (2024) explores the potential of LLMs as predictors or enhancers of graph-based predictions. Other methods, like LLaGA (Chen et al., 2024a) and GraphGPT (Tang et al., 2024), use instruction tuning to map graph data into the LLM embedding space. Similarly, Graphtext (Zhao et al., 2023) and Unigraph (He & Hooi, 2024) adopt NLP techniques, with Graphtext (Zhao et al., 2023) translating graphs into natural language via a syntax tree encapsulating node attributes and inter-node relationships, and Unigraph (He & Hooi, 2024) learning a unified graph tokenizer in a self-supervised fashion to generalize across different attribute domains. Prodigy (Huang et al., 2023) further encodes textual attributes with an LLM and leverages prompt-based graph representations for task generalization.

In contrast, recent approaches forgo LLMs entirely. For instance, Xia & Huang (2024) employs projections of the attribute matrix to handle shifts to new datasets. We exclude this method along with two similar recent approaches (Yu et al., 2024; Zhao et al., 2024a) from our baseline comparisons. This is because these methods' end-to-end architectures for cross-domain graph pre-training employ SVD-based dimensionality reduction primarily as an input processing step, similar to our projection method for unified input space. Given our focus on understanding the effectiveness of input space unification techniques, we study this component in isolation (*c.f.* the baseline *raw* in Section 4) rather than comparing against their full architectures which include additional mechanisms like domain tokens and coordinators. Lachi et al. (2024) employs a Perceiver-based encoder to compress domain-specific attributes into a shared latent space. However, since their method requires finetuning when adapting to unseen out-of-distribution datasets, it falls outside our focus on zero-shot generalization capabilities without additional training. Zhao et al. (2024b) proposes GraphAny, specifically designed for node classification, which models inference on new graphs as an analytical solution to LinearGNNs, and addresses generalization by learning attention scores to fuse predictions from multiple LinearGNNs. However, STAGE outperforms GraphAny in Table 2. Mao et al. (2023) introduces the concept of attribute proximity as a key factor in determining the likelihood of links forming between nodes. Unfortunately, the definition of proximity still depends on the attribute space, making the method unsuitable in our settings of interest. Frasca et al. (2024) proposes "Feature-Structuralization" which converts categorical node features into additional nodes and edges in the graph structure itself. However, adapting this technique would require modification to handle continuous node attributes, making it not directly applicable as a baseline for our work.

Another line of work addresses zero-shot domain transferability on heterogeneous graphs such as knowledge graphs, where both the nodes (entities) and edge types (relations) may be new and unseen on the test-time graph. For instance, ISDEA+ (Gao et al., 2023) proposes a set aggregation layer over the set of edge-type-specific graph representations to ensure equivariance to edge type permutations. Gao et al. (2023) also proposes a theoretical framework named double

equivariance that underlies the necessary design principles of models capable of tackling such a task. In contrast, the theoretical framework of our work addresses transferability to unseen attribute domains and proposes a novel connection between statistical tests and the graph regression task. ULTRA (Galkin et al., 2024) and TRIX (Zhang et al., 2024b), on the other hand, build a relation graph that captures the interactions among different edge types, and apply pipelines based on NBFNet (Zhu et al., 2021c) to ensure equivariance to edge type permutations. Similarly, InGram (Lee et al., 2023) also builds a relation graph, but its relation graph differs from ULTRA's in that it computes a set of affinity scores between pairs of relations and use them as edge weights on the relation graph. In comparison, the STAGE-edge-graphs built by our method captures the statistical dependencies among different attribute dimensions of node attributes in the graph. However, all of these methods rely sorely on graph structure and disregard attributes in nodes. In contrast, our work focus on attributed graphs, which is capable of leveraging important information carried in the node attributes.

Finally, recently Bevilacqua et al. (2025) introduced HoloGNN, a framework that learns node representations transferable across diverse graph tasks. However, HoloGNN assumes a fixed attribute domain and does not address the challenge of generalizing to unseen datasets with differing attribute spaces, which is the focus of our setting.

**Maximal Invariants and Statistical Testing.** Bell (1964) first explored the relationship between invariant and almost-invariant tests in hypothesis testing. Berk & Bickel (1968) and Berk (1970) extended Bell's approach to show that almost-invariant tests are equivalent to invariant ones under certain conditions, which are those met in our work. Later, Berk et al. (1996) explored the interplay between sufficiency and invariance in hypothesis testing by providing counterexamples that demonstrate how these concepts can differ significantly in other scenarios. Recently, Koning & Hemerik (2024) improved the efficiency of hypothesis testing under invariances for large transformation groups such as rotation or sign-flipping without resorting to sampling.

