# OpenReview forum: "Zero-Shot Generalization of GNNs over Distinct Attribute Domains"
_ICML.cc/2025/Conference — ICML 2025 poster_

### Official Review · Reviewer_aT2i · 2025-03-11

**Overall Recommendation:** 4

**Summary:**

The authors study the problem of generalizing GNNs to new graphs that have distinct node/edge attributes. This is a very important problem when attempting to create graph foundation models, as different graphs will often have very different attributes. These attributes will differ not only in dimension size, but in semantic meaning as well. The authors propose a new technique, STAGE, to handle this problem. It first works by constructing a new graph for each node pair (u, v) based on the node attributes (i.e., a "STAGE graph"). Each node corresponds to one attribute and the graph is fully connected. The "node attributes" in the STAGE graph consider the pdf between node attributes. A GNN is run on the STAGE graph for each edge to produce an edge embedding. A second GNN is then run on the original graph where the edge weights are the edge-level embeddings from the previous stage (original attributes are discarded). The authors study the generalization ability of STAGE, showing that it can generalize to unseen node attributes better than baselines.

**Claims And Evidence:**

Overall I believe the claims are well supported by evidence. Theoretically, they show that STAGE has good transferability potential. They further include experiments that seem to demonstrate empirically.

However, currently the authors only show that can transfer models within **graph domains**. Note that this is distinct from attribute domains. For example, all the E-commerce datasets are E-commerce graphs (with the same being said for H&M). Furthermore, Pokec and Friendster are both social networks. As such, even though they contain distinct attributes there is still overlap in the semantic meaning of many node attributes.

This is perfectly fine, however I think it's worth emphasizing that the current results don't show the ability of STAGE to transfer across domains (e.g., train on E-commerce and test on a social network). If possible, I encourage the authors to add such results.

**Essential References Not Discussed:**

None.

**Experimental Designs Or Analyses:**

Most of the experiments are well-designed with good analyses.

However, I think it can be improved in a few instances:

1. The authors argue that STAGE is the only method to consistently increase in performance when considering more graphs (Figure 4). However, this doesn't seem to be true, as NBFNet-Gaussian still increases. The authors mention that NBFNet-Gaussian seems to plateau after 3 graphs. However, the same can be said for STAGE. In reality, the mean performance of STAGE when going from 3 to 4 graphs is barely different.
2. In Table 1 the authors only show the performance when holding out E-Commerce Store and H&M. It would be better if they can also include the raw results when holding out the other 4 E-Commerce datasets. It's fine if this is in the Appendix. I think it would be good to see how consistent this observation is across different hold-out datasets.

**Methods And Evaluation Criteria:**

I think the GNNs used for STAGE and the baselines compared against are suitable. While other baselines exist, the authors choose those that are most common. Furthermore, the evaluation criteria is well aligned with common practice.

**Other Comments Or Suggestions:**

1. I think some more intuitive explanation can be given for why STAGE considers the conditional probabilities of different node attribute types between nodes. It becomes clearer after reading Section 3. However, when reading Section 2, it's unclear why STAGE is designed in this manner. It may help if the authors can provide a brief intuitive explanation early in Section 2 to better motivate this design before jumping into the details.

**Other Strengths And Weaknesses:**

I include a few other weaknesses below:

1. I think the current efficiency results (Tables 9 and 10) downplay the inefficiency of STAGE. As noted, STAGE is quadratic with regards to the number of node attributes (during graph construction and STAGE-edge inference). The current results only show a slight increase in runtime. However, this is mainly due to the fact that the current graphs tend to have very few node attributes (at most 16). However for graphs with more node attributes, STAGE will quickly become impractical to run. This can be seen by the fact that the authors reduce the number of attributes in the original Friendster dataset (644) to a much smaller number.

   This isn't inherently a bad thing. However, this severely limits the real-world potential of STAGE, as it can only handle graphs with few attributes. For example, we can look at the OGB datasets [1], which are common graph datasets. ogbn-arxiv, which is used for node classification has 128 node features. For link prediction datasets, ogbl-collab and ogbl-citation2 have 128 features while ogbl-ppa has 58. It would be impractical to run STAGE on any of these methods (irrespective of graph size).

2. Currently STAGE requires separating attributes into those that are ordered and unordered. As such, some manual pre-processing must be done beforehand. In fact, for some datasets it may not even be known whether some attributes are truely ordered or unordered. This can further be quite tedious when many node attributes exist. However, this is a smaller weakness as it only needs to be done once.

[1] Hu, Weihua, et al. "Open graph benchmark: Datasets for machine learning on graphs." Advances in neural information processing systems 33 (2020): 22118-22133.

**Questions For Authors:**

1. Have you tested the ability of STAGE to generalize across *graph domains* (e.g., from E-commerce and social networks)? If not, would it be possible to test this using additional datasets? To be clear, this experiment isn't *necessary*. However, I think if STAGE can show such transferability, it would greatly enhance the utility of the method. As currently, the authors only show that STAGE can generalize across *attribute domains* among graphs in the same domain (e.g., E-Commerce).

2. Would you be able to include the results when holding out each of the 5 E-Commerce dataset?

**Relation To Broader Scientific Literature:**

This paper is well-situated in the current field of Graph ML. A big impediment to the creation of graph foundation models is that different graphs can contain wildly different node features/attributes. As such, a method that can help alleviate this problem is highly sought. As such, I believe this paper should hold great interest to many in the field of Graph ML.

**Theoretical Claims:**

I read them, however, I didn't look at them in detail.

---

> ### Author Rebuttal · Authors · 2025-04-01
>
> We appreciate the reviewer for recognizing that our paper “should hold great interest to many in the field of Graph ML” and that STAGE can help alleviate the significant “impediment to the creation of graph foundation models.” We now address their remarks.
>
> **Q1.** “There is still overlap in the semantic meaning of many node attributes.”
>
> **A1:** Our Ecommerce dataset comprises five product categories with semantically different attributes. Even when attributes appear semantically similar, they represent fundamentally different attribute domains with different values and distributions. For instance,
>
> - Bed material can be Wood, Metal, Composite
> - Shoe materials can be Leather, Synthetic, Canvas
>
> This highlights STAGE’s ability to transfer across attribute domains that do not overlap in the semantic meaning, making it well-suited for diverse real-world applications.
>
> **Q2.** “Have you tested the ability of STAGE to generalize across graph domains (e.g., from E-commerce and social networks)? If not, would it be possible to test this using additional datasets? To be clear, this experiment isn't necessary.”
>
> **A2:** Our E-commerce datasets focus on temporal link prediction, while the social networks involve node classification. The social network data lacks edge creation times, so we cannot perform the same link prediction task. Additionally, the social network classification task is binary, while the E-commerce data does not have a directly comparable binary task. That said, we share the reviewer’s curiosity, and if they have suggestions for a suitable shared task on a different dataset, we would be happy to explore them.
>
> **Q3.** “The authors mention that NBFNet-Gaussian seems to plateau after 3 graphs. However, the same can be said for STAGE. In reality, the mean performance of STAGE when going from 3 to 4 graphs is barely different.”
>
> **A3:** Thank you, we will revise our statement to be more precise. While not the only method showing improvement, STAGE demonstrates particularly favorable scaling properties with additional graph domains. Specifically, STAGE exhibits notably tighter interquartile ranges compared to NBFNet-Gaussian at higher domain counts, suggesting more reliable performance across different domain combinations. Additionally, STAGE's lower whiskers consistently rise with more domains, showing that even its worst-case scenarios improve with more training data - a pattern less pronounced in NBFNet-Gaussian.
>
> **Q4.** “Would you be able to include the results when holding out each of the 5 E-Commerce dataset?”
>
> **A4:** We will include them in the revision.
>
> **Q5.** “STAGE is quadratic with regards to the number of node attributes” “This severely limits the real-world potential of STAGE, as it can only handle graphs with few attributes.”
>
> **A5:** While the quadratic complexity may be perceived as a limitation, we argue that it does not substantially diminish the usefulness of STAGE. In fact, numerous popular machine learning algorithms exhibit similar complexity constraints, including the Transformer Attention mechanism, which has a quadratic constraint. In relational deep learning, methods designed for small data thrive and make huge impacts on the real world. A recent example is TabPFN [1], a foundation model for tabular data that has found extensive scientific and business applications in real-world settings despite being designed for small datasets. Certain eigen-decomposition-based GNNs also have quadratic or cubic complexity.
>
> [1] Hollmann et al., 2025. Accurate predictions on small data with a tabular foundation model
>
> **Q6.** “STAGE requires separating attributes into those that are ordered and unordered. As such, some manual pre-processing must be done beforehand.” “This can further be quite tedious when many node attributes exist. However, this is a smaller weakness as it only needs to be done once.”
>
> **A6:** We agree that this step may require upfront effort. However, it's worth noting that for categorical attributes, this processing can be done in lieu of mapping them into one-hot encodings, which is a common practice in ML pipelines. Additionally, for unordered attributes that are not categorical, it is likely that a pipeline already exists and can be repurposed.
>
> **Q7.** “Some more intuitive explanation can be given for why STAGE considers the conditional probabilities of different node attribute types between nodes.”
>
> **A7:** Thank you, we will include it in the revision. The key intuition is that zero-shot generalization requires focusing on statistical relationships between attributes rather than absolute values. By modeling conditional probabilities, STAGE recognizes patterns across different attribute spaces. For instance, instead of learning specific rules like 'phones with 8GB RAM tend to be expensive,' STAGE learns abstract relationships such as 'high values in X_1 correlate with high values in X_2,' enabling knowledge transfer across domains with different attributes.

---

> > ### Comment · Reviewer_aT2i · 2025-04-03
> >
> > Thank you for the clarifications and promise of revisions. I think they will help strengthen the paper. I'll keep my positive score.
> >
> > > That said, we share the reviewer’s curiosity, and if they have suggestions for a suitable shared task on a different dataset, we would be happy to explore them.
> >
> > My apologies, but I can't think of any dataset that has a suitable number of node features for STAGE (most common Graph ML datasets tend to have >100 node features). Unrelated to this review, I think it would be interesting if the authors can find such a dataset. If STAGE can generalize across such datasets, that would be quite noteworthy.

---

> > > ### Author Response · Authors · 2025-04-03
> > >
> > > Thank you for your feedback and for maintaining your positive score. We appreciate your curiosity and are actively exploring datasets that could demonstrate generalization from e-commerce to social networks. If we identify appropriate ones, we will make sure to include them in the revision.

---

### Official Review · Reviewer_3CxP · 2025-03-14

**Overall Recommendation:** 3

**Summary:**

The paper introduces STAGE, a novel framework designed to overcome the challenge that traditional GNNs face when node attributes in test graphs differ from those seen during training. Rather than relying on raw attribute values, STAGE computes statistical dependencies between pairs of attributes by constructing a dedicated “STAGE-edge-graph” for each edge. A two-stage GNN approach is then applied: one GNN extracts edge embeddings from these graphs, and a second GNN uses these embeddings (while discarding the original node attributes) to produce the final graph representation. The method is underpinned by theoretical analysis and is validated through experiments across diverse datasets.

## Update after Rebuttal

Most of my concerns have been addressed, and I will maintain my current score.

**Claims And Evidence:**

The paper’s main claims—that STAGE can achieve zero-shot generalization by learning domain-independent statistical dependencies between node attributes—are supported by both rigorous theory and comprehensive experiments. The theoretical claims (e.g., Theorems 3.2–3.4) are backed by detailed proofs (in Appendix B) and the experimental results show significant improvements (up to 103% gain in Hits@1 for link prediction and improved node classification accuracy) compared to baselines. One minor concern is that the evidence is demonstrated on small to medium-sized datasets; thus, the scalability and robustness on very large graphs remain somewhat unverified.

**Essential References Not Discussed:**

No.

**Experimental Designs Or Analyses:**

Yes, I checked the soundness of the experiment section.

**Methods And Evaluation Criteria:**

The proposed method—constructing STAGE-edge graphs that capture pairwise statistical dependencies and using a two-stage GNN framework—is innovative and well aligned with the problem of handling distinct attribute domains. Overall, the methodological choices and evaluation setup are sensible, though the quadratic complexity for the number of attributes may limit its applicability in larger-scale settings.

**Other Comments Or Suggestions:**

No.

**Other Strengths And Weaknesses:**

This paper’s strength lies in its innovative STAGE framework, which unifies heterogeneous node attributes via statistical dependencies backed by rigorous theory and extensive empirical validation across diverse datasets. For the weaknesses:
1. The approach is quadratic in complexity concerning the number of attributes, which may limit its applicability to very large graphs or datasets with high-dimensional attribute spaces.
2. While experiments on small to medium-sized datasets are promising, how well STAGE scales and performs on much larger real-world graphs remains to be seen.

**Questions For Authors:**

1. Could the authors provide an empirical comparison or toy visual example that highlights how order statistics capture invariant relationships better than normalized raw values in Sec. 3.1? How does the use of order statistics handle outliers or tied values in attribute distributions?

2. "Then, by dropping the attribute identifiers in STAGE-edge-graphs, we sacrifice maximal expressivity but ensure that STAGE is invariant to permutations of the attribute order. "Could the authors provide more theoretical or experimental illustration on how this loss of identifiers affects model expressiveness, especially in cases where attribute order might carry implicit information?

3. Could the authors discuss how sensitive STAGE is to the diversity of training domains theoretically or experimentally?

4. The limitation of STAGE seems not to be discussed in the manuscript.

**Relation To Broader Scientific Literature:**

The paper is well-contextualized within the literature on graph neural networks, domain adaptation, and zero-shot learning.

**Theoretical Claims:**

I examined the theoretical contributions, particularly the proofs related to Theorems 3.2, 3.3, and 3.4. These proofs appear rigorous and well-founded.

---

> ### Author Rebuttal · Authors · 2025-04-01
>
> We appreciate the reviewer for recognizing our theory is “rigorous and well-founded.” We now address their comments.
>
> **Q1.** “The approach is quadratic in complexity concerning the number of attributes, which may limit its applicability to very large graphs or datasets with high-dimensional attribute spaces.”
>
> **A1:** While the quadratic complexity may be perceived as a limitation, we argue that it does not substantially diminish the usefulness of STAGE. In fact, numerous popular machine learning algorithms exhibit similar complexity constraints, including the Transformer Attention mechanism, which has a quadratic constraint. In the domain of relational deep learning, methods designed for small data thrive and make huge impacts on the real world. A recent example is TabPFN [1], a foundation model for tabular data that has found extensive scientific and business applications in real-world settings despite being designed for small datasets. Certain eigen-decomposition-based GNNs also have quadratic or cubic complexity.
>
> [1] Hollmann et al., 2025. Accurate predictions on small data with a tabular foundation model
>
> **Q2.** “Could the authors provide an empirical comparison or toy visual example that highlights how order statistics capture invariant relationships better than normalized raw values in Sec. 3.1? How does the use of order statistics handle outliers or tied values in attribute distributions?”
>
> **A2:** Thanks for the suggestion. Next, we present a toy example adapted from the Ecommerce dataset to illustrate why order statistics are superior to normalization for domain transfer.
>
> Domain 1 (Train - Computers have attribute power_supply_watts):
>
> - Computer A: 800 W
> - Computer B: 600 W
> - Computer C: 450 W
> - Computer D: 300 W
>
> Domain 2 (Test - Refrigerators have attribute energy_rating):
>
> - Refrigerator A: 4.0 [A]
> - Refrigerator B: 3.0 [B]
> - Refrigerator C: 2.0 [C]
> - Refrigerator D: 1.0 [D]
>
> Now assume there exists a correspondence between Computer A and Refrigerator A, Computer B and Refrigerator B, and so on. This correspondence arises because users who prefer high-powered computers tend to select refrigerators with better energy ratings. When encoding these attributes, we need a representation that preserves this user preference pattern across domains, despite the different attribute scales.
>
> With z-score normalization, the values become different across domains:
>
> - Product A: 1.42 vs 1.34
> - Product B: 0.34 vs 0.45
> - Product C: -0.47 vs -0.45
> - Product D: -1.28 vs -1.34
>
> With order statistics, namely STAGE, *the values remain identical* in both domains:
>
> - Product A: 0.25 (1/4 products)
> - Product B: 0.5 (2/4)
> - Product C: 0.75 (3/4)
> - Product D: 1.0 (4/4)
>
> Therefore, while normalization produces different values for the corresponding products, order statistics preserve consistency across domains by capturing ranking information, which is invariant to monotonic transformations. Additionally, by focusing on ranks, order statistics are inherently robust to outliers, and ties are handled by assigning the average rank to all tied items.
>
> **Q3.** “Could the authors provide more theoretical or experimental illustration on how this loss of identifiers affects model expressiveness, especially in cases where attribute order might carry implicit information?”
>
> **A3:** Without attribute identifiers, the model may lose the ability to distinguish attributes. For instance, consider two edges e1 and e2 in the original graph, and two features f1 and f2. If f1 has CDF value v1 in e1 and v2 in e2, while f2 has CDF value v2 in e1 and v1 in e2, with all other features having identical CDFs in both edges, then the two STAGE edge-graphs become isomorphic. In this case, STAGE produces identical edge embeddings for e1 and e2, even though the actual attribute distributions differ, limiting the model's expressivity and ability to distinguish them. However, our empirical analysis suggests this theoretical limitation has minimal practical impact. As shown in Figure 6, STAGE effectively captures meaningful attribute relationships even without explicit identifiers.
>
> **Q4.** “Could the authors discuss how sensitive STAGE is to the diversity of training domains theoretically or experimentally?”
>
> **A4:** Figure 4 shows that STAGE benefits from diverse training domains. STAGE’s performance consistently improves as we increase the number of training domains, with higher median values and tighter confidence intervals for both Hits@1 and MRR metrics. This suggests that STAGE effectively leverages the diversity in multiple graph domains to learn more robust and transferable representations. In contrast, the other baselines show minimal or inconsistent improvements with additional domains.
>
> **Q5.** “The limitation of STAGE seems not to be discussed in the manuscript.”
>
> **A5:** We acknowledge the limitations of STAGE on larger graphs in Sections 1 and 4. We appreciate the reviewer's suggestion and will further clarify them in the revision.

---

### Official Review · Reviewer_7ofD · 2025-03-14

**Overall Recommendation:** 3

**Summary:**

This paper studies the zero-shot generalization of GNNs under the shift in attribute domains. They propose the STAGE algorithm that aims to model the statistical dependencies between node attributes that can be invariant across domains instead of the original node attribute. Specifically, STAGE creates the edge graph for each edge that models the conditional probabilities between attribute pairs from each edge end points. Then, they use one GNN to generate edge embeddings based on edge graph probabilities and another GNN to output the embeddings for task. The paper also includes theoretical justification in terms of the expressiveness and transferability of STAGE. Lastly, the experiments on various node classification and link prediction tasks show the effectiveness of STAGE.

**Claims And Evidence:**

**Strength:**
- Investigating the change of attribute domains can be useful in many real world adaptation scenarios
- The idea of statistical dependencies between attributes rather than raw features are interesting and novel

**Weakness / Questions:**
- This paper only focuses on the shift in attribute domains, but in reality it is rather rare that the graphs only shift in terms of the node attribute domains. How do you view the effectiveness of this method under the structure shift and will STAGE become problematic under the presence of structure shift?
- The paper claims to work only for the small to medium datasets with controlled number of attributes, which limit the practical usage of the algorithm.

**Essential References Not Discussed:**

It can be more comprehensive if you have a more in-depth discussions on graph generalization under distribution shifts, like graph OOD. Also, the works included in the discussion are rather outdated. Specifically, you can discuss the similarity and difference in the shifts considered and how your method compared to previous literatures.

**Experimental Designs Or Analyses:**

**Strength:**
- The empirical improvements are significant
- The datasets span different types of tasks and domains

**Weakness:**
- limited size of datasets as mentioned above
- The baselines are rather simple and lack of potential related baselines. Even if this paper targets specifically attribute domain shift, it might still be interesting to compare to foundation model that claims to have zero-shot generalization ability. Also, is it possible to compare with graph OOD works in similar settings.

**Methods And Evaluation Criteria:**

**Strength:**
- Interesting idea and clear explanation in the methodology section.
- The design is not restricted a specific type of shift in attribute domains and consider some detailed setting like unordered and ordered types of attributes.
- Can be applicable to different types of tasks, e.g node classification and link prediction.

**Weakness / Questions:**
- The complexity of creating edge graphs still remain concerning and limit the usage to large datasets
- The current design only considers the pairwise relations, which is a rather simplified version of the attribute hypergraph

**Other Comments Or Suggestions:**

N/A

**Other Strengths And Weaknesses:**

Please refer to the above sections.

**Questions For Authors:**

Please refer to the above sections.

**Relation To Broader Scientific Literature:**

The key contribution is the idea of statistical dependencies that might can be transferable under the change of attribute domains.

**Theoretical Claims:**

**Strength:**
- Detailed and extensive theoretical motivation of STAGE

**Question:**
- Potentially can have a better connection to the actual design of the STAGE algorithm more. For instance, what design corresponds to "assigning unique attribute identifiers to label the nodes of our STAGE-edge-graphs" and what design corresponds to "dropping the attribute identifiers".

---

> ### Author Rebuttal · Authors · 2025-04-01
>
> We appreciate the reviewer’s recognition that STAGE “can be useful in many real-world adaptation scenarios,” and thank them for acknowledging STAGE’s versatility, as well as for their appreciation of our theoretical motivation and experiments. We now address their comments.
>
> **Q1:** “..It might still be interesting to compare to foundation models that claim to have zero-shot generalization ability.”
>
> **A1:** We already include GraphAny in Table 2, a baseline claimed to have zero-shot generalization ability. Per your suggestion, we include another foundation model, namely GCOPE [1] in Table 2 (reporting zero-shot test accuracy on Pokec, trained on Friendster), and report the results below. Notably, **STAGE outperforms the GCOPE foundation model by 21.9%.**
>
> | | **Accuracy** (↑) | **% gain** |
> |--|--|--|
> | GraphAny | 0.591 ± 0.0083 | 10.3% |
> | GCOPE | 0.535 ± 0.0153 | 21.9% |
> | STAGE | 0.652 ± 0.0042 | 0% |
>
> **Q2:** “How do you view the effectiveness of this method under the structure shift and will STAGE become problematic under the presence of structure shift?” And “Is it possible to compare with graph OOD works in the same settings?”
>
> **A2:** Our experiments account for substantial structural variation, as shown in Table 3 (Appendix C), where the number of nodes, edges, and average degrees often double from training to testing. Despite this, STAGE consistently outperforms other baselines in all zero-shot settings (Figure 3), demonstrating its effectiveness in scenarios where both the attribute domain and the structure shift.
>
> Nonetheless, we acknowledge that STAGE does not make explicit assumptions about structural shifts, but it also does not impose constraints on them, and can be integrated with any GNN supporting input edge embeddings. We anticipate that integrating STAGE with existing graph-OOD GNNs, which are explicitly designed for structural shifts (e.g., varying graph sizes), could further enhance performance in these settings.
>
> Finally, to the best of our knowledge, existing graph OOD methods (e.g., those in [2]), focus on structural shifts but do not address the attribute space shifts we consider, which include changes in the number of attributes between train and test. As a result, **existing graph OOD methods cannot be evaluated in our setting**. We will clarify this distinction in our revision and include related works on graph OOD.
>
> **Q3:** “The complexity of creating edge graphs still remains concerning and limits the usage on large datasets.”
>
> **A3:** The computational complexity of STAGE is linear in the number of edges and quadratic in the number of attributes, rendering it particularly suitable for small-to-medium datasets. While this characteristic may be perceived as a limitation, we argue that it does not substantially diminish the usefulness of STAGE. In fact, numerous popular machine learning algorithms exhibit similar complexity constraints, including the Transformer Attention mechanism, which has a quadratic constraint. In the domain of relational deep learning, methods designed for small data thrive and make huge impacts on the real world.  A recent example is TabPFN [3], a foundation model for tabular data that has found extensive scientific and business applications in real-world settings despite being designed for small datasets. Certain eigen-decomposition-based GNNs also have quadratic or cubic complexity.
>
> **Q4:** “The current design only considers the pairwise relations, which is a rather simplified version of the attribute hypergraph”
>
> **A4:** Our STAGE-edge-graph captures marginal probabilities (e.g., P(A), P(B), P(C)) and pairwise conditional probabilities (e.g., P(A|B)) among attributes of node pairs. However, we agree that this representation requires additional assumptions about independence to recover joint probabilities of three or more attributes. While extending STAGE to incorporate these higher-order interactions is theoretically possible, it would result in increased complexity, whose exploration represents an important research direction on its own.
>
> **Q5:** “What design corresponds to assigning unique attribute identifiers to label the nodes of our STAGE-edge-graphs and what design corresponds to dropping the attribute identifiers?”
>
> **A5:** In a STAGE-edge-graph, labeling each node with the attribute ID of an endpoint node corresponds to assigning unique attribute identifiers. In contrast, our design of representing each node solely by its probability density function, without the attribute ID, corresponds to dropping the attribute identifiers. We will incorporate it in the next revision, thank you.
>
> [1] Zhao et al., 2024. All in One and One for All: A Simple yet Effective Method towards Cross-domain Graph Pretraining
>
> [2] Zhang et al., 2025. A Survey of Deep Graph Learning under Distribution Shifts: from Graph Out-of-Distribution Generalization to Adaptation
>
> [3] Hollmann et al., 2025. Accurate predictions on small data with a tabular foundation model

---

> > ### Comment · Reviewer_7ofD · 2025-04-03
> >
> > Thank you for the rebuttal, it addressed some of my questions so I raise my score to 3.

---

> > > ### Author Response · Authors · 2025-04-03
> > >
> > > We appreciate your consideration of our rebuttal and the increased score. In our revised manuscript, we will incorporate the additional GCOPE foundation model comparison, clarify the distinction between our approach and graph OOD methods, and elaborate on the technical details of our STAGE-edge-graph implementation as promised.

---

### Official Review · Reviewer_fV8y · 2025-03-14

**Overall Recommendation:** 3

**Summary:**

The paper introduces STAGE (Statistical Transfer for Attributed Graph Embeddings) to enhance the zero-shot generalization capabilities of Graph Neural Networks (GNNs). It represents node attributes using order statistics, treating node features as random variables and reconstructing them into a STAGE-edge-graph based on the probability density functions between the attributes of two nodes connected by an edge. The method employs a two-stage GNN to obtain edge embeddings and graph representations, capturing statistical information while ignoring numerical attribute values. This approach enables zero-shot generalization across datasets with different categories, names, semantics, and cardinalities. Experiments focus on small-to-medium-sized GNNs, aiming to test generalization across different datasets within the same domain.

**Claims And Evidence:**

The claims made in the paper are supported by referenced literature and experimental data.

**Essential References Not Discussed:**

The paper does not omit any essential related references.

**Experimental Designs Or Analyses:**

The main experiment focuses on link prediction, training on the E-Commerce Stores dataset and testing on untrained domains (e.g., training on bed and desk domains, testing on refrigerators and smartphones) and the H&M Personalized Fashion Recommendations dataset. Comparisons are made against various baseline methods (linear mapping, Gaussian noise, pure structural modeling, textual modeling, normalized features, and supervised structural modeling) on a unified GNN (NBFNet), measuring Hits@1 and MRR metrics, which is reasonable.

**Methods And Evaluation Criteria:**

The STAGE method improves the generalization of GNNs across different datasets within the same domain (e.g., e-commerce, product recommendation), namely there is a shift in feature distributions between training and test sets, but the task objectives remain the same. Compared to baseline models, STAGE shows significant improvements. The evaluation criteria involve prediction Hits@1 and MRR metrics for link prediction and node prediction tasks, providing convincing evidence.

**Other Comments Or Suggestions:**

In Section 3.1, the first paragraph, "e.g., R^d for d ≥ 1, where the total order ≤ is well defined" should have a letter after the ≤ symbol.

**Other Strengths And Weaknesses:**

The core of the STAGE method involves modeling and reconstructing each edge in the original graph into a STAGE-edge-graph, which increases computational overhead, limiting experiments to small-to-medium-sized graphs and preventing scalability to larger graphs.

The modeling method requires graph attributes to be continuous values or discrete categories, so the main experiments focus on the e-commerce domain. However, its effectiveness on text attribute graphs is very limited, as shown in Appendix C3, where STAGE performs no better than other baseline methods on social network graphs.

**Questions For Authors:**

No questions at present.

**Relation To Broader Scientific Literature:**

The proposed STAGE method models graph data from a non-parametric statistical perspective, seeking invariant statistical properties of graphs to improve cross-domain generalization of GNNs, marking a significant step towards foundational graph models.

**Theoretical Claims:**

The paper models graph features using order statistics from statistics, with detailed theoretical derivations that do not appear to contain errors.

---

> ### Author Rebuttal · Authors · 2025-04-01
>
> We appreciate the reviewer’s recognition of STAGE as *“marking a significant step towards foundational graph models”* and their positive assessment of our theory and empirical results. We will incorporate their suggestions into the revised manuscript. Below, we address their remarks:
>
> **Q1.** Building each edge a STAGE-edge-graph “increases computational overhead, limiting experiments to small-to-medium-sized graphs and preventing scalability to larger graphs.”
>
> **A1:** The computational complexity of STAGE is linear with respect to the number of edges and quadratic with respect to the number of attributes, rendering it particularly suitable for small-to-medium datasets. While this characteristic may be perceived as a limitation, we argue that it does not substantially diminish the usefulness of STAGE. In fact, numerous popular machine learning algorithms exhibit similar complexity constraints, including the Transformer Attention mechanism, which has a quadratic constraint. In the domain of relational deep learning, methods designed for small data thrive and make huge impacts on the real world.  A recent example is TabPFN [1], a foundation model for tabular data that has found extensive scientific and business applications in real-world settings despite being designed for small datasets. Certain eigen-decomposition-based GNNs also have quadratic or cubic complexity.
>
> The effectiveness of STAGE is evident in our experiments, for instance on the Ecommerce Stores dataset, where zero-shot learning achieves a Hits@1 score of nearly 0.6 when predicting user behavior from desktop configurations described by just 12 attributes. This result underscores that the computational trade-off enables substantial performance gains in settings where computational complexity is less of a constraint than model quality and expressiveness.
>
> [1] Hollmann et al., Nature 2025. Accurate predictions on small data with a tabular foundation model
>
> **Q2.** STAGE’s “effectiveness on text attribute graphs is very limited, as shown in Appendix C3, where STAGE performs no better than other baseline methods on social network graphs.”
>
> **A2:** We believe the reviewer is referring to Table 5, discussed in Appendix E, which presents an additional experiment focusing on zero-shot prediction of *age* on social networks. As explained in Appendix E, this experiment was designed to highlight the inherent difficulty of predicting age in a zero-shot setting, given that "age" and "gender" are the only shared attributes available for node labels. Our goal was to illustrate that age prediction is fundamentally challenging across models, justifying our focus on gender prediction (Table 2). Consistent with this, neither our STAGE nor text embeddings outperformed other approaches in age prediction. All models struggled due to the significant distributional shift between age attributes in the Friendster and Pokec networks, as visualized in Figure 5.
>
> Nonetheless, we agree with the reviewer that STAGE is designed to handle non-text attributes, theoretically guaranteeing generalization when considering continuous and discrete, as well as ordered and unordered attributes. We believe that extending STAGE to handle text-attribute, for instance by coupling it with an initial text encoder, represents an interesting avenue for future research, although one that requires a separate effort.
>
> **Q3.** Minor suggestions
>
> **A3:** We thank the reviewer’s suggestion on manuscript change and will include it in the updated version.

---

### Decision · Program_Chairs · 2025-05-01

**Decision:**

Accept (poster)

**Comment:**

This paper presents STAGE, a novel framework for enabling zero-shot generalization of GNNs across distinct node attribute domains. By modeling statistical dependencies between node attributes using order statistics and a two-stage GNN pipeline, STAGE effectively abstracts away from raw attribute values, allowing transfer across domains with mismatched attribute sets. The paper is well-motivated, theoretically grounded, and supported by strong empirical results on both link prediction and node classification tasks. Reviewers agree that the proposed approach addresses a relevant and challenging problem in graph learning. While concerns remain about STAGE's scalability due to its quadratic complexity with respect to the number of attributes, the authors provide a compelling case that the method is effective and practically useful for small to medium-scale graphs. The rebuttal thoroughly addresses reviewer questions, offering additional comparisons and clarifying theoretical design choices. Overall, this is a solid contribution that opens up a promising direction for domain-agnostic graph representation learning, and I recommend acceptance.